# The composition and structure of the outer kinetochore KMN complex is conserved across kingdoms
Dipesh Kumar Singh [1,2] ✉, Birgit Walkemeier[1], Anjali Nayini[2], Jelle Van Leene[3,4], Stéphanie Durand[1], Geert De Jaeger [3,4], Raphael Guerois [5] & Raphael Mercier [1] ✉

In eukaryotes, chromosome segregation relies on attachment to the spindle, ensured by the kinetochore. The outer kinetochore attaches to the microtubules and is named after three sub-complexes KNL1C, MIS12C, and NDC80C (KMN). While the KMN complex comprises ten proteins in humans *S. cerevisiae*, its conservation in more distant eukaryotes is unclear. Here, we aimed to define the KMN complex in the plant Arabidopsis using affinity purification and identified thirteen KMN proteins. Seven were previously known to have a conserved function (atMIS12, atNNF1, atNDC80, atSPC24, atSPC25, atNUF2, and atKNL1) and six were uncharacterized. These six proteins show remote similarity to yeast/human KMN-associated proteins, whose homologs have not yet been characterized in plants. We named them atDSN1, atCSM1, atNSL1.1/.2, and atZWINT1.1/.2. We confirmed kinetochore localization for atDSN1, atCSM1, atNSL1.1, and atZWINT1.1 *in planta*. In addition, *atDSN1*, *atCSM1*, and *ZWINT1.1/.2* are essential, further supporting their kinetochore function. AlphaFold3 predicts an alike3D organization of the KMN complex in plants and mammals. We conclude that the KMN complex is globally conserved with a matching composition and similar organization in distant eukaryotes, with some local variations, suggesting its presence in the common ancestors of all living eukaryotes.

Kinetochores are large multisubunit protein complexes attached to the centromeric chromatin of chromosomes. Kinetochores connect with the positive ends of microtubules at metaphase and drag chromatids to cellular poles during cell division. Most of the constitutive proteins and regulatory proteins of the kinetochore are conserved between *Saccharomyces cerevisiae* and mammals indicating that their basic architecture is retained[1–4]. The kinetochore is divided into two main parts: the inner and outer kinetochore[5,6]. The inner kinetochore interacts with centromeric chromatin whereas the outer kinetochore interacts with the spindle[7]. The loading of the inner kinetochore begins through the histone variant CENH3, which is crucial for assembling the kinetochore complex at the centromeric region of chromosomes[8–10]. CENPC, interacts with CENH3 via its C-terminus[8,11] and with its N-terminus interacts with the outer kinetochore KMN (Knl1, Mis12, Ndc80) complex[6,12].

The KMN complex comprises ten subunits well-conserved from yeast to Human and divided into three subcomplexes: the KNL1 complex (Knl1,

Zwint), the MIS12 complex (Mis12, Dsn1, Nsl1, Nnf1), and the NDC80 complex (Ndc80, Nuf2, Spc24, Spc25)[13,14]. Several KMN subunits have alternate names in different species. For example, Nnf1 is known as PMF1 in humans[15,16] Mis12 is referred to as Mtw1[17,18] and ZWINT1 is called Kre28 in budding yeast[19]. *Saccharomyces cerevisiae* has an additional subunit in the Mis12 subcomplex, Csm1, which appears to be absent in mammals. Csm1 contributes to chromosome segregation in both meiosis[20] and mitosis[21]. The KMN complex promotes the assembly of numerous auxiliary microtubule-associated components, microtubule regulatory proteins (BUB protein), and Spindle assembly checkpoint (SAC) regulatory proteins[22–27]. Mis12 is a central kinetochore protein that interacts with all subunits of the Mis12 complex (Dsn1, Nsl1, Nnf1) and the N-terminus of CENPC[28]. It also interacts with the C-terminal globular domain of Knl1 and the SPC24-SPC25 dimer.

Concerning the plant clade, seven subunits of the KMN complex have been functionally characterized in Arabidopsis: MIS12, NNF1, SPC24, SPC25, NUF2, NDC80, and KNL1[29–34]. These subunits localize to the

[1]Department of Chromosome Biology, Max Planck Institute for Plant Breeding Research, Cologne, Germany. [2]Laboratory of Plant Reproduction, Center for DNA Fingerprinting and Diagnostics (CDFD), Hyderabad, India. [3]Department of Plant Biotechnology and Bioinformatics, Ghent University, Ghent, Belgium. [4]Center for Plant Systems Biology, VIB, Ghent, Belgium. [5]Institute for Integrative Biology of the Cell (I2BC), Commissariat à l'Energie Atomique, CNRS, Université Paris-Sud, Université Paris-Saclay, Gif-sur-Yvette, France. ✉e-mail: dipesh@cdfd.org.in; mercier@mpipz.mpg.de

kinetochore and are essential, as homozygous null mutants are inviable and heterozygous plants exhibit one-quarter embryo lethality in developing seeds[29,30,32–34]. Some of the KMN subunits such as KNL1, MIS12, and NDC80 have also been identified as kinetochore components in other plants, including maize, *Physcomitrella patens*, and *Cuscuta*[35–38]. Additionally, potential homologs of four other yeast and human kinetochore subunits DSN1, CSM1, NSL1, and ZWINT1 have been suggested to be present in plants based on limited sequence similarity. However, these homologs have not been experimentally confirmed as kinetochore components nor functionally characterized[4,20,35,39,40].

Here, we define the plant KMN complex by using two potential subunits (CSM1 and DSN1) and the recently characterized subunit KNL1 as bait in affinity purification coupled to mass spectrometry (AP-MS) experiments in Arabidopsis, combined with *in planta* localization, mutant analysis, and structure predictions. We conclude that the Arabidopsis KMN complex presents a strikingly conserved composition and organization to its yeast/mammalian counterpart, with each of the eleven subunits having its counterpart in Arabidopsis. This suggests that the 11-subunits-KMN complex was present before the lineages of plant and fungi/animals separated, and thus probably in the common ancestor of all present eukaryotes.

## Results

### Affinity purification coupled to mass spectrometry Identify KMN proteins

We aimed to characterize the Arabidopsis KMN complex, and notably to identify all its components. We selected three proteins based on sequence similarity to represent the Arabidopsis homologs DSN1 (AT3G27520), CSM1 (AT3G20070), and KNL1 (AT2G04235). DSN1 and CSM1 were suggested to be present in plants but have not been functionally characterized, while KNL1 was recently characterized[4,20,30,31,36,38]. We used these three proteins as baits for protein complex analysis by affinity purification coupled to mass spectrometry (AP-MS) in Arabidopsis cell cultures[41]. We performed five independent AP-MS experiments in triplicate, covering N- and C-terminal GS^rhino fusions of CSM1 and DSN1, whereas for KNL1 the N-terminal fusion was analyzed. Data are available via ProteomeXchange with identifier PXD062209. To identify specific interactors in each triplicate AP-MS experiment, non-specific proteins were removed by quantitatively comparing normalized spectral abundance factor (NSAF) values of all identified proteins against a corresponding large control AP-MS dataset covering non-related bait proteins (see methods). After filtering out false positives (see methods), a total of 42 specific potential interactors were identified in at least one of the AP-MS experiments (Table 1 and Fig. 1). Strikingly, 19 proteins were robustly identified in at least three out of five experiments, with 16 of them being systematically identified in all five AP-MS experiments (Table 1, Fig. 1). These include the baits DSN1, CSM1, and KNL1, each of which was identified in its own AP-MS experiment as was technically expected, but also each time with the two other bait proteins, showing that DSN1, CSM1 and KNL1 interact with each other in vivo. In addition, all the six previously known subunits of the KMN complex (NDC80, NUF2, SPC24, SCP25, MIS12 and NNF1)[29,32–34] and two additional kinetochore proteins, CENPC and BMF1 were co-purified in all five AP-MS experiments (Table 1, 2, Fig. 1)[42–44]. CENPC connects the inner kinetochore with the KMN complex, while BMF1 interacts with the KMN complex at the spindle[42–44]. These results clearly confirm that AtKNL1, AtDSN1, and AtCSM1 belong to the KMN complex.

Four additional proteins candidate to be KMN components were very robustly identified in the AP-MS experiments: AT3G23910, AT3G24255, AT4G00525, and AT1G01715 (Table 1, Fig. 1 and Table 2). These correspond to two pairs of paralogs, as AT3G23910 and AT3G24255 are 95% identical (Figs. S1 and S2) and AT4G00525 and AT1G01715 share 43% identity (Figs. S1 and S3). Homology search using BLAST or PSI-BLAST identified homologs of these proteins in the plant kingdom but not in other eukaryote clades[45]. However, remote homology searches using HHpred with sensitive parameters (E-value cutoff 1e-3; max target hits 250; iterations 3; minimum probability 20%; minimum coverage 20%)[46,47] identified the

KMN subunit ZWINT1 as the top hit among human proteins for AT3G23910/AT3G24255 and NSL1 as the top hit for AT4G00525/AT1G01715. This is consistent with the previous suggestion based on distant sequence similarities that ZWINT1 and NSL1 homologs are present in plants[4,20,36]. We designate AT3G23910/AT3G24255 as ZWINT1.1/ZWINT1.2, and AT4G00525/AT1G01715 as NSL1.1/NSL1.2.

All five AP-MS experiments consistently detected both ZWINT1.1 and ZWINT1.2 (Table 1, 2), which share 95% sequence identity. In total, we identified 20 peptides corresponding to common sequences of ZWINT1.1 and ZWINT1.2, contributing to a total peptide count of 894. We also identified five peptides that are specific to ZWINT1.1. and five specific to ZWINT1.2. The unique peptides' total counts were 123 for ZWINT1.1 and 100 for ZWINT1.2 (Figs. S4A, and Supplementary Dataset 1). This demonstrates that both paralogs are present in the kinetochore and indicates that ZWINT1.1 is slightly more abundant than ZWINT1.2.

NLS1.1 and NSL1.2 have 45% identity, and all identified peptides in the AP-MS experiments were unambiguously attributed to one of the two paralogs. NSL1.1 was consistently detected in all five AP-MS experiments, whereas NSL1.2 was identified in only three (with DSN1 and KNL1 baits, but not with CSM1) (Table 1, 2). In total, we identified three unique peptides for both NSL1.1 and NSL1.2, which yielded 128 peptide counts for NSL1.1 and 22 for NSL1.2 across (Supplementary Dataset 1 and Fig. S4B). This demonstrates that both NSL1.1 and NSL1.2 are present at the kinetochore, with NSL1 being ~fivefold more abundant than NSL1.2 at the kinetochore in Arabidopsis cell cultures.

We also found peptides, though less robustly, from two other kinetochore-associated proteins, BUB3.3[43,44] and AT3G10180 (Table 1). BUB3.3 is involved in the kinetochore as a key regulator of chromosome segregation during cell division, ensuring proper attachment of spindle[23,27]. AT3G10180 shows remote homology with human CENP-E through HHpred and Foldseek analysis[48,49], has also been suggested to be present in Arabidopsis[4] and was recently reported as KAK1, kinetochore kinesin protein 7[44]. CENP-E is a kinetochore motor protein crucial for the stable, bi-oriented attachment of chromosomes to the spindle[50,51]. We also robustly identified ESD4 (AT4G15880) in all five AP-MS. ESD4 is a SUMO protease associated with early flowering and reduced growth when mutated[52,53]. The association of ESD4 with CSM1, DSN1, and KNL1 reported here, suggests a potential, unexplored role for ESD4 in kinetochore dynamics and chromosome segregation, akin to the role of SUMO protease SENP6 in animals[54]. Twenty-three additional proteins were identified in the AP-MS assays, showing relatively low NSAF ratios compared to the core KMN subunits (Table 1). Despite their lower abundance, these proteins were retained using stringent filtering criteria, and might thus represent genuine interactors that may have a weaker or more transient association with the kinetochore and are good candidates for future functional analysis.

In brief, the series of AP-MS experiments showed that the proposed Arabidopsis DSN1, CSM1, KNL1, ZWINT1.1/.2, and NLS1.1/.2 proteins robustly associate with the kinetochore, and in particular with the KMN complex. Next, we aimed to explore their function *in planta*.

### DSN1, CSM1, KNL1 ZWINT1.1 and NSL1.1 co-localize with CENH3

The AP-MS assays and remote similarity described above suggest the identification of so-far uncharacterized subunits of the Arabidopsis KMN complex, CSM1, DSN1, NSL1.1/.2, and ZWINT1.1/.2. To test their predicted localization to the kinetochore, we tagged DSN1, KNL1, NSL1.1, ZWINT1.1, BUB3.3, and GUS as negative control with the fluorescent protein Venus (variant of the Green Fluorescent Protein GFP)[55] at the N-terminus under the ubiquitin10 promoter and transformed plants. We used three transformed plants per tagged protein for localization analysis. Immunolocalization were performed with anti-GFP antibodies on fixed tissue from unopened floral buds to detect the tagged proteins. CSM1 localization was studied using an anti-CSM1 polyclonal antibody raised against a specific peptide (see Materials and Methods). DNA was stained with DAPI and the centromeres were marked through immunolocalization

**Table 1 | AP-MS of DSN1, CSM1 and KNL1**

| Bait used for the AP-MS | | Number of positive replicates out of three (Affinity Enrichment Score) | | | | |
|---|---|---|---|---|---|---|
| Gene name | Protein name or annotation | DSN1-CGSrhino | NGSrhino-DSN1 | CSM1-CGSrhino | NGSrhino-CSM1 | NGSrhino-KNL1 |
| AT3G27520 | DSN1 | 3 (1043351) | 3 (754465) | 3 (39715) | 3 (107130) | 3 (187473) |
| AT1G61000 | NUF2 | 3 (499278) | 3 (390055) | 3 (43527) | 3 (112944) | 3 (214776) |
| AT3G08880 | MUN/SPC24 | 3 (311870) | 3 (255567) | 3 (24466) | 3 (77961) | 3 (114722) |
| AT4G19350 | NNF1 | 3 (254225) | 3 (174948) | 3 (16377) | 3 (47765) | 3 (91088) |
| AT2G04235 | KNL1 | 3 (220379) | 3 (158551) | 3 (15598) | 3 (65815) | 3 (327207) |
| AT3G54630 | NDC80 | 3 (213395) | 3 (164535) | 3 (24209) | 3 (60327) | 3 (95407) |
| AT3G48210 | SPC25 | 3 (204965) | 3 (138528) | 3 (12493) | 3 (46621) | 3 (76219) |
| AT4G00525 | NSL1.1 | 3 (177567) | 3 (134245) | 3 (28048) | 3 (58279) | 3 (112060) |
| AT5G35520 | MIS12 | 3 (167854) | 3 (89767) | 3 (12223) | 3 (27378) | 3 (60705) |
| AT3G20070 | CSM1/TITAN9 | 3 (121865) | 3 (75657) | 3 (592365) | 3 (539461) | 3 (50632) |
| AT1G01715 | NSL1.2 | 3 (35770) | 3 (36498) | | | 2 (3484) |
| AT2G20635 | BUB1MAD3/BMF1 | 3 (32217) | 3 (22074) | 3 (5278) | 3 (19112) | 3 (8702) |
| AT3G24255 | ZWINT1.2 | 3 (21723) | 3 (14788) | 3 (1173) | 3 (4111) | 3 (23006) |
| AT3G23910 | ZWINT1.1 | 3 (7926) | 3 (5317) | 3 (371) | 3 (1273) | 3 (9801) |
| AT1G15660 | CENPC | 3 (8745) | 3 (6291) | 3 (2013) | 3 (7043) | 3 (2401) |
| AT4G15880 | ESD4 | 3 (14583) | 3 (9354) | 3 (10052) | 3 (13348) | 3 (4206) |
| AT1G19980 | - | 3 (28223) | 3 (25236) | 3 (8622) | 3 (15805) | 3 (12207) |
| AT1G69400 | BUB3,3 | 3 (13541) | 3 (4436) | | | 3 (15773) |
| AT3G10180 | CENP-E | 3 (3337) | 3 (473) | | | 3 (2653) |
| AT3G57000 | - | | 3 (119) | | 3 (76) | |
| AT2G22230 | Thioesterase superfamily protein | 3 (107) | | | | |
| AT1G17720 | Protein phosphatase 2 A | 2 (81) | | | | |
| AT3G24530 | AAA-type ATPase family | | 3 (90) | | 3 (149) | |
| AT5G08600 | U3 ribonucleoprotein (Utp) family | | 3 (167) | | 3 (94) | |
| AT4G10960 | UGE5 | | 2 (264) | | | |
| AT5G65770 | CRWN4 | 3 (189) | | | | |
| AT3G10530 | Transducin/WD40 repeat-like superfamily | | | 3 (260) | 3 (127) | |
| AT2G07707 | Mitochondrial ATPase | | | 3 (738) | | |
| AT5G43720 | rRNA-processing EFG1-like | | | 3 (663) | | |
| AT2G29500 | HSP20-like chaperones superfamily | | | | | 3 (2751) |
| AT4G22740 | glycine-rich protein | | | | | 3 (2917) |
| AT1G03050 | AtECA2/PICALM5a | 3 (6377) | 3 (4258) | | | |
| AT2G44680 | CKB4|casein kinase II beta subunit 4 | 2 (1446) | | | | |
| AT5G24630 | BIN4, MID|double-stranded DNA binding protein | | | | | 3 (4560) |
| AT5G25190 | ESE3|Integrase-type DNA-binding superfamily protein | | | | | 2 (1467) |
| AT1G25260 | Ribosomal protein L10 family protein | | | | | 3 (376) |
| AT5G15550 | atPEIP2, AtPEP2|Transducin/WD40 repeat-like superfamily protein | | | | | 3 (216) |
| AT1G13160 | ARM repeat superfamily protein | | | | | 3 (196) |
| AT5G50310 | Galactose oxidase/kelch repeat superfamily protein | | | | | 3 (185) |
| AT4G28250 | ATEXPB3, ATHEXP BETA 1.6, EXPB3|expansin B3 | | | | | 3 (132) |
| AT5G06460 | ATUBA2, UBA 2|ubiquitin activating enzyme 2 | | | | | 3 (122) |
| AT1G50480 | THFS|10-formyltetrahydrofolate synthetase | | | | | 3 (121) |

Five AP-MS were performed, each one with three independent purifications followed by mass spectrometry. After filtering, the specifically enriched proteins are reported, with their times identified in three experiments and their enrichment score (=the product of the NSAF ratio and significance ($-\log(p$ value)) vs the large dataset. Data are available via ProteomeXchange with identifier PXD062209.

of CENH3, a variant of histone H3 specifically localized to the centromeric chromatin[56]. Remarkably, for each of the tested proteins, CSM1, GFP-DSN1, GFP-KNL1, GFP-NSL1.1, and GFP-ZWINT1.1 (Fig. 2A–E) the signal appears as 10 foci in most observed cells, matching the expected number of kinetochores in Arabidopsis somatic cells. Some cells displayed 5 to 9 foci, likely due to closely positioned kinetochores merging into a single focus. Further, the CSM1, DSN1, KNL1, NSL1.1, and ZWINT1.1, foci systematically colocalized with the CENH3 foci (Fig. 2O–S). We also found

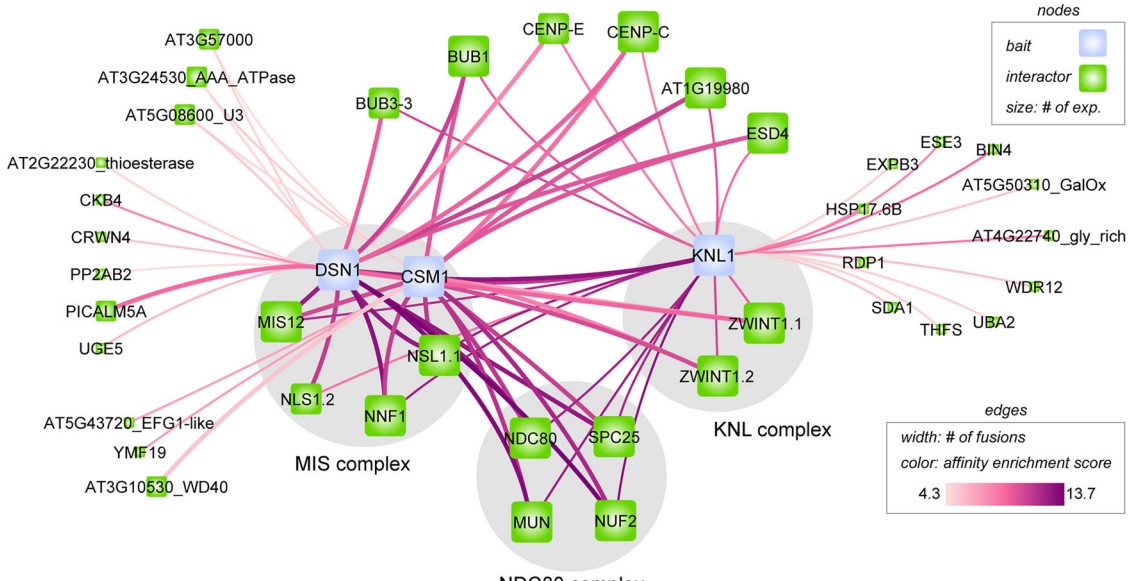

**Fig. 1 | AP-MS of DSN1, CSM1 and KNL1.** The blue nodes represent the baits and the green nodes the identified interactors (Table 1). The node size represents the number of experiments in which an interactor was found in (out of the 5 AP-MS). The edge width indicates if an interaction was found with both fusions (N- and C-GSrhino) or only one fusion (N- or C-GSrhino). The edged are colored according to the Ln-transformed Affinity Enrichment Scores. Generated with Cytoscape v3.10.2 and edited with Illustrator.

that all GFP/CENH3 foci colocalized within intense DAPI signals (Fig. 2V–Z). Brighter DAPI staining indicates a region of tightly packed DNA typically associated with centromeric heterochromatin. In contrast, GFP-BUB3.3 (Fig. 2F) on interphase somatic cells did not appear as a detectable signal above background levels, and in particular, no GFP-BUB3.3 signal was detected at CENH3 foci (Fig. 2M), consistent with previous reports[43], and indicating that BUB3.3 is not a constitutive component of the kinetochore. Concerning the negative control GFP-GUS, we detected GUS activity in the plant tissue (Figure S5), confirming the expression of the GFP-GUS fusion protein, but no GFP signal was detected to colocalize with CENH3 (Fig. 2U). These results demonstrate that CSM1, DSN1, KNL1, NSL1.1, and ZWINT1.1 localize to the kinetochores of interphase cells *in planta*, further supporting their proposed role as KMN subunits. Note that independent recent studies showed that CSM1, DSN1, KNL1, NSL1.1, and ZWINT1.1 localize to the kinetochore throughout the mitotic cell cycle[57,58]. We also showed that CSM1 localizes to kinetochore at mitosis and meiosis (Fig. 3). On mitotic metaphase chromosomes, CSM1 localized to the outer region of the kinetochore, oriented to the spindle pole, and CENH3 localized to the inner kinetochore region (Fig. 3). This spatial arrangement is consistent with CSM1 being a member of the outer kinetochore KMN complex.

## Most KMN genes are essential

Previous studies have demonstrated that null mutations in KMN subunits (MIS12, NNF1, NDC80, NUF2, SPC24, SPC25) result in embryo lethality, with ~one-quarter of aborted seeds in the fruits of heterozygous plants (expected Mendelian segregation for selfing) and absence of homozygous in their progeny (Table 2)[29,32–34]. This highlights the critical importance of each KMN complex subunit for proper cell division. *CSM1, DSN1,* and *KNL1* are single-copy genes, and we characterized two independent mutant alleles, either isolated from public collections of T-DNA insertions or generated through CRISPR-CAS9 targeted mutagenesis (Fig. S1). The plants heterozygous for each mutation (*knl1, csm1* or *dsn1*) were indistinguishable from the wild-type in terms of growth and development. However, their fruits contained approximately 25% shrunken seeds (Fig. 4 and Supplementary Dataset 2). This observation is consistent with the expected Mendelian segregation ratio of 3:1 (Chi-square tests, *p*-values of 0.80 for *knl1*, 0.30 for *csm1*, and 0.86 for *dsn1*), indicating that the quarter of homozygous embryos are not viable. In the progeny of these heterozygous plants, we

observed only heterozygous and wild-type plants, with a complete absence of homozygous mutants in each case. The observed segregation ratios were similar: *csm1-1* (96 heterozygous / 30 wild type), *csm1-2* (90/21), *dsn1-1* (95/22), *dsn1-2* (85/32), *knl1-1* (147/38), and *knl1-2* (95/24). This shows that Arabidopsis *CSM1, DSN1* and *KNL1* are essential genes, which is compatible with a central role in the kinetochore.

The *ZWINT1* and *NSL1* both have two paralogs in the Arabidopsis genome and the four encoded proteins were detected in the AP-MS experiments described above, indicating the presence of each of them in the kinetochore. *NSL1* genes are located on different chromosomes (chromosomes 1 and 4), while the *ZWINT1* paralogs are nearby, 150 kb apart, on chromosome 3. We validated their expression and coding sequences through cDNA amplification and sequencing (Supplementary Dataset 3 and Fig. S1).

Notably, *ZWINT1.2* contains a 1.4 kilobase transposon in the third intron (not present in ZWINT1.1) which does not appear to affect splicing. The AT3G24255.2 gene prediction in the current annotation of the Arabidopsis database (TAIR10) matches our *ZWINT1.2* analyzed cDNA sequence. ZWINT1.1 and ZWINT1.2 thus encode proteins of similar size with 95% identity (Fig. S2). We generated three independent CRISPR mutant lines with large or complete deletion of the *ZWINT1.1* coding sequence (Fig. S1 and Supplementary Dataset 3). Plants heterozygous for each *zwint1.1* mutation appear indistinguishable from wild type in terms of growth and fertility, a notably do not show any shrunken seeds in their fruits (Fig. 4). However, when we grew the progeny of heterozygous plants in soil, we could not obtain homozygous mutants but only heterozygous and wild-type plants (heterozygous/wild type: *zwint1.1-1* 72/16, *zwint1.1-2* 80/25, *zwint1.1-3* 72/32). When the seeds from heterozygous plants were sown in vitro, we observed plantlets dying shortly after germination for the three *zwint1.1* alleles (Fig. S6A), with a frequency compatible with 1:4 (dead/viable plants: *zwint1.1-1* 17/79, *zwint1.1-2* 21/78, *zwint1.1-3* 20/76). Genotyping of dying plantlets in the progeny of *zwint1.1-1* heterozygous plants confirmed that they were all homozygous for the *zwint1.1-1* mutation (*n* = 15/15). This indicates that *ZWINT1.1* is an essential gene, like other KMN complex subunits. However, *zwint1.1* embryos appear to be able to develop until the germination stage, perhaps due to a partial compensation by ZWINT1.2, but fail to further develop.

Concerning *ZWINT1.2*, we analyzed two T-DNA alleles, and a series of complete deletions (Fig. S1). The two T-DNA alleles, *zwint1.2-1* and *zwint1.2-2*, do not appear to affect plant viability, as we observed, mendelian

**Table 2 | Comprehensive summary of Arabidopsis thaliana outer kinetochore**

| | Arabidopsis gene ID | Arabidopsis Protein | Purified with the bait: | | | | | localisation at interphase | Homolog in mammals | Homolog in S. Cerevisiae | Mutant phenotype | reported in Arabidopsis |
|---|---|---|---|---|---|---|---|---|---|---|---|---|
| | | | DSN1 Cter | DSN1 Nter | CSM1 Cter | CSM1 Nter | KNL1 Nter | | | | | |
| MIS12 complex | AT3G27520 | DSN1 | x | x | x | x | x | centromeric | DSN1 | Dns1 | lethal | this study |
| | AT3G20070 | CSM1, TTN9 | x | x | x | x | x | centromeric | not found | Csm1 | lethal | this study. Zhang et al.[58] |
| | AT4G00525 | NSL1.1 | x | x | x | x | x | centromeric | NSL1 | Nsl1 | slight fertility reduction | this study |
| | AT1G01715 | NSL1.2 | x | x | o | o | x | not tested | | | | this study |
| | AT4G19350 | NNF1 | x | x | x | x | x | centromeric | Nnf1 (PMF1) | Nnf1 | lethal | Allipra et al.[29] |
| | AT5G35520 | ATMIS12 | x | x | x | x | x | centromeric | MIS12 | mtw1 | lethal | Sato et al.[33] |
| NDC80 complex | AT3G54630 | NDC80, HEC1 | x | x | x | x | x | centromeric | HEC1/Ndc80 | Ndc80 | lethal | Shin et al.[34] |
| | AT3G48210 | SPC25 | x | x | x | x | x | centromeric | SPC25 | Spc25 | lethal | Shin et al.[34] |
| | AT3G08880 | MUN | x | x | x | x | x | centromeric | MUN1/Spc24 | Spc24 | lethal | Shin et al.[34] |
| | AT1G61000 | NUF2 | x | x | x | x | x | centromeric | NUF2 | nuf2 | lethal | Li et al.[32] |
| KNL complexs | AT2G04235 | KNL1 | x | x | x | x | x | centromeric | KNL1 | Spc105 | lethal | this study/ Deng et al.[30] He et al.[31] |
| | AT3G23910 | ZWINT1.1 | x | x | x | x | x | centromeric | ZWINT1 | Kre28 | lethal | this study |
| | AT3G24255 | ZWINT1.2 | x | x | x | x | x | not tested | | | wild-type like | this study |
| | AT2G20635 | BMF1 | x | x | x | x | x | centromeric | Bub1/Mad3 | Bub1/Mad3 | wild-type like | Komaki & Schnittger[43] |
| | AT1G15660 | CENPC | x | x | x | x | x | centromeric | CENPC | Mif2 | lethal | Ogura et al.[42] |
| | AT3G10180 | CENPE | x | x | o | o | x | centromeric | CENPE | not found | wild-type like | Tang et al.[44] |
| | AT1G69400 | BUB3.3 | x | x | o | o | x | not detected | BUB3 | Bub3/Pac9 | wild-type like | Komaki & Schnittger[43] |
| | AT4G15880 | ESD4 | x | x | x | x | x | nuclear membrane | | | weak plant | Murtas et al.[53] |
| | AT1G19980 | – | x | x | x | x | x | not tested | | | not tested | |
| | AT3G57000 | – | x | x | o | x | o | not tested | | | not tested | |

**Fig. 2 | KMN subunits co-localize with CENH3.**
Immunolocalization of CENH3 and GFP using anti-CENH3 and anti-GFP antibodies in interphase somatic cells. **A** anti-CSM1, **B–G** anti-GFP, **H–N** anti-CENH3, **O** colocalization of CSM1 with CENH3, (**P–U**) colocalization of GFP with CENH3, (**V**) merge of CSM1, CENH3 and DAPI, (**W–Z**, AA, AB) merge of GFP, CENH3 and DAPI. Scale bar = 2 μm.

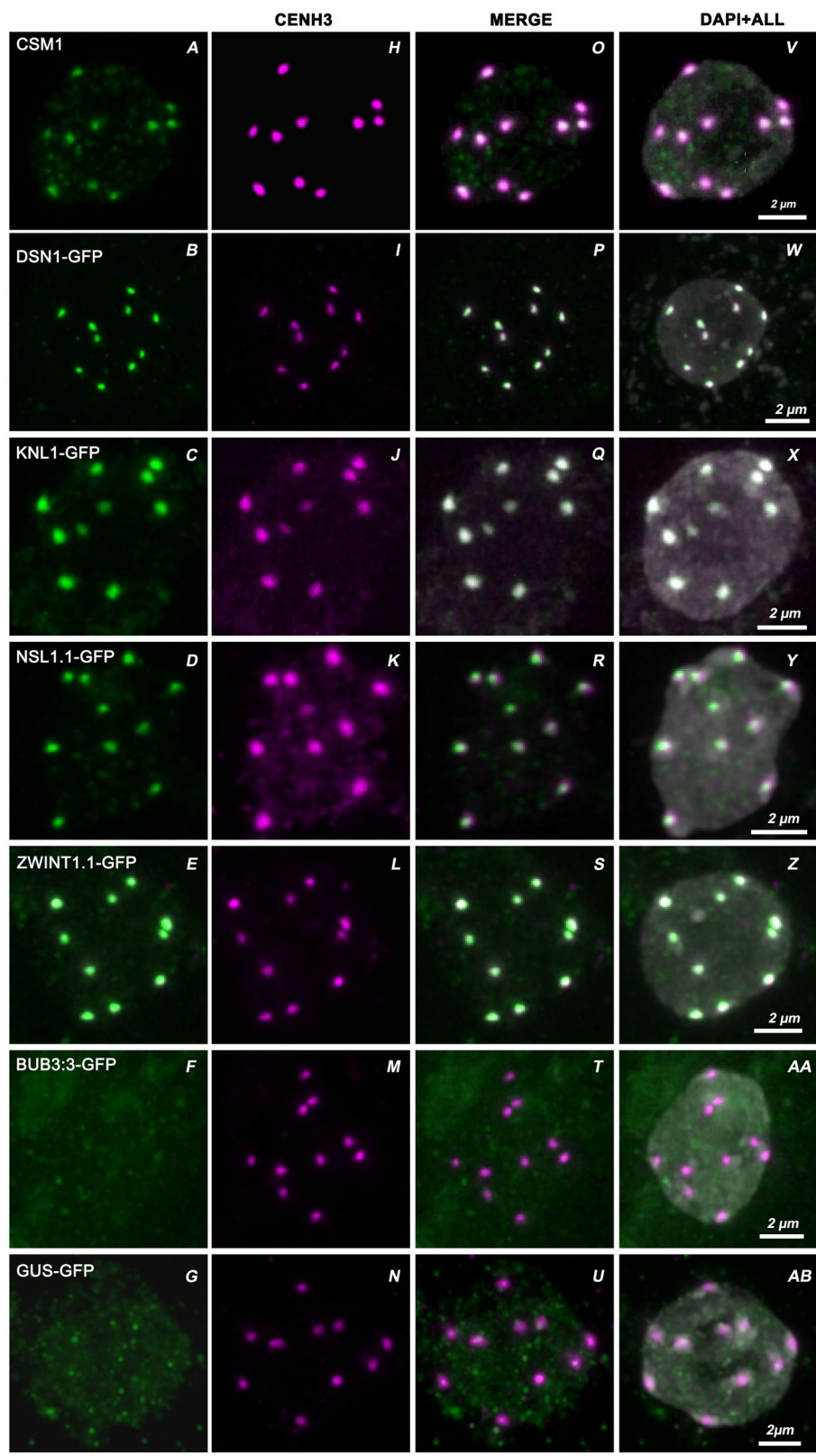

segregation in the progeny of heterozygous plants (Wild types/Hetero-zygotes/mutants *zwint1.2-1* 26/45/21; *zwint1.2-2* 22/41/19), along with normal growth and fertility of homozygous mutants (Fig. 4). However, for both alleles the T-DNA insertion is in the third intron, and while no splicing of the T-DNA was observed, mRNA expression at wild-type levels was detected downstream of the T-DNAs (Fig. S7). The expressed mRNA may encode a truncated protein using an alternative start codon (e.g., M135). We

then generated four independent mutants of *ZWINT1.2* using CRISPR (derived from four T1 plants). The four alleles, *zwint1.2-3, zwint1.2-4, zwint1.2-5 and zwint1.2-6* have an identical full deletion of the *ZWINT1.2* gene (Fig. S1 and Supplementary Dataset 3). In the progeny of each het-erozygous mutants, we observed only heterozygous and wild-type plants, with a complete absence of homozygous mutants (heterozygous/wild type: 78/41, 67/46, 90/50 and 75/36, respectively). We also observed plantlets

**Fig. 3 | Immunolocalization of CSM1 and CENH3 in mitotic and meiotic cells. A–D** Mitotic metaphases with Immunolocalization of CSM1 (red) and CENH3 (green). The DNA is stained with DAPI (gray). CENH3 marks the inner kinetochore associated with chromatin, and CSM1 colocalize with CENH3, with a shift toward the cellular poles. This supports the conclusion that CSM1 belongs to the outer kinetochore. Scale bar = 1 μm. **E–H** Meiotic cell at pachytene, with immunolocalization of CSM1 (red), CENH3 (green) and the cohesin subunit REC8 (blue). REC8 decorates the pair of axis of the synapsed chromosomes. CENH3 marks the kinetochore regions and CSM1 colocalize with it. Scale bar = 3 μm. **I–L** Meiotic cell at late anaphase I with Immunolocalization of CSM1 (red) and CENH3 (green). The DNA is stained with DAPI (gray). CSM1 colocalizes with CENH3 at the kinetochore Scale bar = 5 μm.

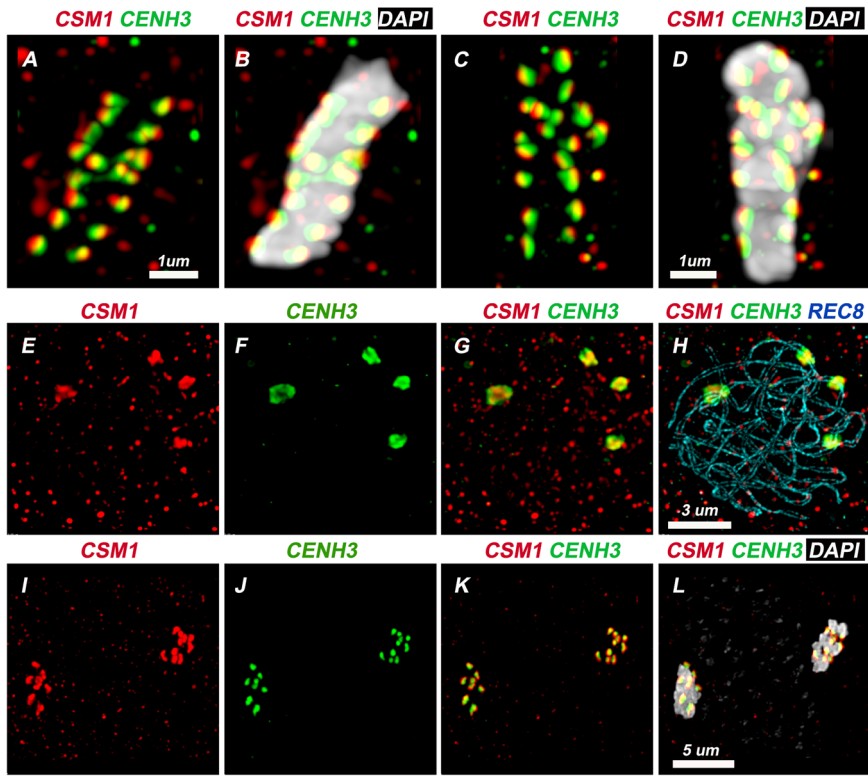

**Fig. 4 | Fertility of KMN mutants. A** Frequency of normal versus aborted seeds per fruit. Fertility was assessed as the average number of seeds per fruit, measured on five plants per genotype and ten fruits per plant. Plum and shrunken seeds were scored as normal/aborted. The wild type of all segregating genotypes were pulled together and represented as a single bar. The value above each bar represents the mean number of normal (top) and aborted (bottom) seeds per fruit ± standard deviation (**B**) representative image of wild-type fruit and (**C**) *dsn1-1* (−/+) fruit, showcasing both aborted and normal seeds.

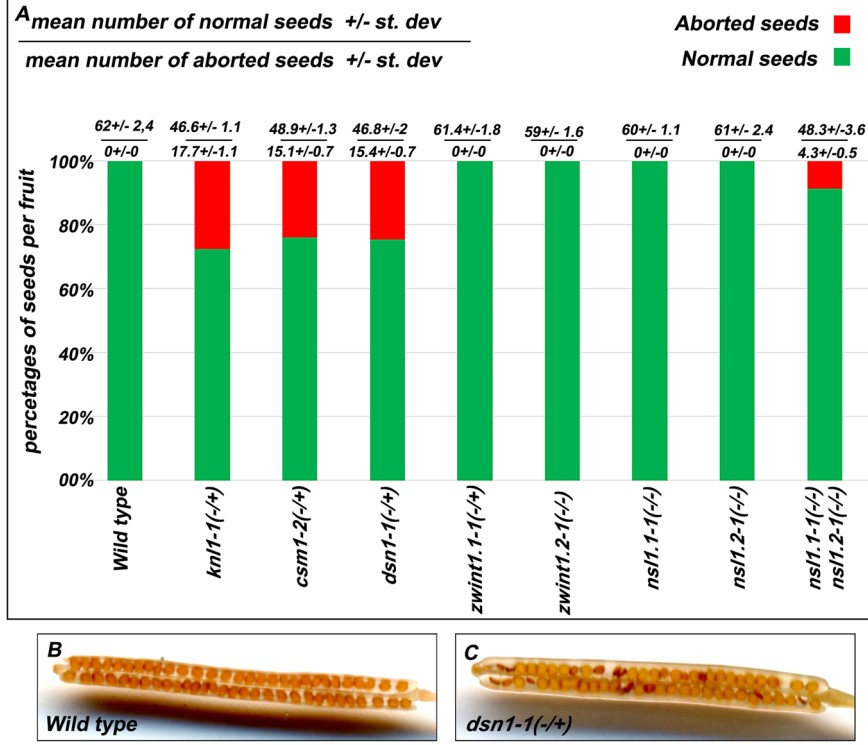

dying shortly after germination (dead/viable plants: *zwint1.2-3*, 21/85, Fig. S6B). This indicates that *ZWINT1.2* is an essential gene and that *zwint1.2-1* and *zwint1.2-2* encodes functional proteins. In summary, both *ZWINT1.1* and *ZWINT1.2* are essential, with a likely partial redundancy allowing development until the germination stage.

Finally, we aimed to functionally characterize the two *NSL1* paralogs. We analyzed two *nsl1.1* mutant alleles, one with a 112 bp deletion in the first

exon (*nsl1.1-1*) and one T-DNA insertion (*nsl1.1-2*) (Fig. S1), and two T-DNA lines for *NSL1.2* (*nsl1.2-1* and *nsl1.2-2*; Fig. S1).

In *nsl1.1-1* RT-PCR with primers flanking the deletion, revealed the expression of a transcript of reduced length and apparently reduced expression. RT-PCR with primers downstream of the deletion similarly indicated diminished expression compared to wild-type (Fig. S8B). Given that the *nsl1.1-1* deletion induces a frameshift at codon 38/140, it is unlikely

that *nsl1.1-1* express a functional protein. The *nsl1.1-2 T-DNA* is inserted in the *NSL1.1* single intron. RT-PCR with primers flanking the intron/TDNA failed to amplify in *nsl1.1-2*, in contrast to wild-type, suggesting that the T-DNA insertion prevent splicing (Fig. S8C). RT-PCR with primers in the second exon revealed transcript levels comparable to those of the wild type, implying the potential presence of an unspliced mRNA or of a truncated transcript with the T-DNA acting as a cryptic promoter (Fig. S8C). In absence of splicing, the *nsl1.1-2* allele is unlikely to yield a functional NSL1.1 protein.

In *nsl1.2-1* the TDNA is inserted at the beginning of the first exon, and RT-PCR with primers flanking the insertion failed to amplify. However, RT-PCR with primers downstream of the insertion detected expression in the mutant, albeit at apparent lower level than in wild type (Fig. S9). This suggests that *nsl1.2-1* expresses a truncated mRNA, possibly capable of initiating translation at methionine-29 (M29/139), resulting in a shortened NSL1.2 protein with unclear functionality. Finally, despite a T-DNA insertion in the 5'UTR, the *nsl1.2-2* allele showed mRNA expression indistinguishable from wild type, and may thus express a functional protein.

When we grew the progeny of *nsl1.1-1*, *nsl1.1-2*, *nsl1.2-1*, and *nsl1.2-2* heterozygous plants in soil and performed genotyping, we obtained heterozygous, wild-type, and homozygous mutant plants in proportions fitting with an unbiased Mendelian segregation (35:18:14 for *nsl1.1-1*, 22:16:12 for *nsl1.1-2*, 50:24:20 for *nsl1.2-1*, and 46:18:20 for *nsl1.2-2*). For each of the four mutations, the obtained homozygous mutants were indistinguishable from wild-type plants in terms of growth, development, and fertility (Fig. 4). We then combined *nsl1.1-1* and *nsl1.2-1* mutations, which both affect the first exon of their respective gene. In the F2 progeny of double heterozygous *nsl1.1-1* (−/+) *nsl1.2-1* (−/+), we obtained 4 double mutants among 96 plants, which was close to the expected number under mendelian segregation (6 expected). The double mutant is viable and does not show major defects though showed a slower growth (Figure S10) and 8.8% seed abortion in fruits (Fig. 4). This suggests that, unlike other KMN subunits, NSL1 is not essential in *Arabidopsis thaliana*. However, as *nsl1.1-1* and *nsl1.2-1* show some mRNA expression, it is quite possible that some functionality was retained in one or both of the mutants. Obtaining full deletion mutations would be required to clarify this question and reach a conclusion on the role of NSL1.1/NSL1.2

### Predicted structural divergence and conservation of the KMN complex across species

The global composite model, generated using AlphaFold3 predictions, provides a structural overview of the outer kinetochore assembly, excluding disordered regions, and reveals its core architecture in *A. thaliana* (Fig. 5A). The global topology is overall similar to that observed in the experimental structures of the human outer kinetochore[13,14].The MIS12 and KNL1 modules are predicted to fold as hetero-tetrameric coiled-coils while the NDC80 module consists of a heterodimeric coiled-coil. For a more detailed inter-species analysis of the KMN assembly, we generated AF3 models for the *H. sapiens* and *S. cerevisiae* complexes. While global similarities are noticeable, significant structural differences exist in Zwint1 and Knl1 proteins. Zwint1 shows significant divergence, featuring two RWD domains in *A. thaliana*, only one in *S. cerevisiae*, and none in *H. sapiens* (Fig. 5B). The absence of the RWD domain in *H. sapiens* is highlighted by a dashed circle in the model (Fig. 5B). In *A. thaliana*, the first RWD domain in the tandem is predicted to interact with the C-terminal helix of Mis12, whereas in *H. sapiens*, the absence of the RWD tandem in ZWINT is associated with a shorter Mis12 homolog. Interestingly in *S. cerevisiae*, a C-terminal extension of Mis12 (Mtw1) is also elongated as in *A. thaliana* but is predicted to interact with the RWD domain of the Knl1 ortholog (Spc105), highlighting the remarkable plasticity of KMN subunit interactions over long evolutionary times. Similarly, models of the MIS12 module demonstrate inter-species variability: *A. thaliana* lacks the Dsn1-Nsl1 four-helix bundle found in *H. sapiens* and *S. cerevisiae* (Fig. 5C), with the dashed circle indicating the absence of a helix bundle). Upstream of this four-helix bundle, the N-terminal end of Dsn1 in *H. sapiens* and *S. cerevisiae* is predicted to interact

with Pmf1 (Nnf1) and Mis12 four-helix bundle. This interaction was shown to play a self-inhibitory effect competitive with CENPC binding[14]. In the predicted *A. thaliana* model of the MIS12 module, the Dsn1 is short and not predicted to interact with Pmf1 and Mis12 as in *H.sapiens* and *S. cerevisiae*.

As for the Csm1 subunit, a homolog of the *S. cerevisiae* Csm1 protein was identified in association with the *A. thaliana* KMN complex, suggesting that it was present before the two lineages diverged. As the plant lineage diverged very early from the fungi/animalia branch in the evolution of eukaryotes, it suggests that Csm1 was present in the common ancestor of all living eukaryotes. In humans, no ortholog of Csm1 has been detected so far, suggesting its loss during evolution after the fungi/animalia splitting. AF3 (Fig. 6) predicts AtCSM1 binds to KMN through the N-terminal extension of Dsn1, similar to *S. cerevisiae*. The atCSM1-DSN1 direct interaction was recently experimentally confirmed[58]. In *A. thaliana*, DSN1 is predicted to fold as a hairpin upon CSM1 binding while it binds as a helix in yeast (from both the experimentally observed and predicted complex), further highlighting the versatility of interaction modes. Finally, we analyzed the interaction between the models of the MIS12 complex module and CENPC. In all three species, a short disordered region in the N-terminal moiety of CENPC is predicted to interact with the Mis12-Pmf1 four-helix bundle (Fig. 6C). For *H. sapiens* and *S. cerevisiae*, the strong agreement between the model and recently solved experimental structures[13,14,20] indicates that the predicted binding mode proposed for *A. thaliana* is also potentially reliable.

### Predicted interactions between atESD4, atBUB3.3, AtBMF1/MAD3 and CENPE with the KMN complex

We explored whether AtESD4, the SUMO protease robustly found in our IP-MS experiments (Table 1) could be predicted to interact with any of the 12 subunits already positioned in the outer kinetochore physical interaction network. We screened AtESD4 against all these potential partners using AlphaFold3 and a fragmentation strategy for non-globular regions[59]. Only for AtCSM1, a significant interaction ipTM score of 0.35 could be reached. The model presented in Fig. 7A suggests that a helical stretch in AtESD4-spanning residues (91-116) could bind the coiled-coil of AtCSM1 with a most likely stoichiometry of 2:2. Interestingly, a recent report also identified AtESD4 as a direct binder of AtCSM1 by both yeast two-hybrid and AP-MS experiments[58] strengthening the relevance of using AlphaFold3 and related AI-based approach to scan physical interactomes for potential direct point of interaction.

In the case of AtBUB3.3, an interaction could be predicted between a small disordered region at the N-terminus of KNL1 spanning two stretches (260–275) and (403–421) (Fig. 7B). Interestingly, the binding mode of the (260–275) stretch folds as an extended motif against the lateral side of the AtBUB3.3 WD40 β-propeller domain, in a manner similar to that observed for the interaction between the yeast homologs of BUB3 and KNL1, namely Bub3 and Spc105[27] [PDB 4BL0]. In yeast, the "Met-Glu-Leu-Thr" (MELT) motif has been emphasized as mediating the interaction upon phosphorylation of the last Thr residue. In *A. thaliana*, the prediction suggests that the corresponding motif in KNL1 would be "Met-Ser-Ile-Val," with the possibility that phosphorylation of the Ser could establish a salt bridge with an Arg residue in BUB3.3.

Regarding the AtBMF1/MAD3 prey, its homologs in yeast and vertebrates, known as Mad3/BubR1, were shown to interact with homologs of BUB3[60,61]. A so-called GLEBS motif in Mad3/BubR1 was shown to mediate the interaction with Bub3. However, this conserved motif, located in the disordered regions downstream of the TPR domain, is not found in *A. thaliana* BMF1. Instead, there is a very short linker between the TPR domain and the pseudo-kinase domain in AtBMF1, preventing the presence of any such disordered interaction motif. Accordingly, while AlphaFold3 unambiguously predicts the existence of the GLEBS motif in human, it does not predict any interaction between AtBUB3 and AtBMF1. We therefore wondered whether another protein in the KMN network could be predicted to interact directly with AtBMF1. Among the list of kinetochore interactors, the only protein predicted to be a potential binder of AtBMF1/MAD3 is AtCENPC, through a small disordered region in its N-terminal extremity

**Fig. 5 | Global model of the KMN complex.**
**A** Global composite model based on AlphaFold3 predictions of the outer kinetochore assembly generated using the protein sequences identified in the proteomic screen and represented by their molecular surfaces and not showing the disordered regions. **B** The comparative analysis focused on the KNL module between the structural models of three species, *A. thaliana*, *H. sapiens* and *S. cerevisiae* generated using AlphaFold3. The folded regions of Zwint and Knl1 proteins are represented as dark and light pink cartoons, respectively. Zwint exhibits the highest degree of structural divergence with either two, one, or no RWD domains in *A. thaliana, S. cerevisiae* and *H. sapiens*, respectively. Absence of RWD domain for human is highlighted by a dashed circle (**C**). Comparative analysis focused on the MIS12 module between the structural models of the same three species as in (**B**). A dashed circle in the (**A**). thaliana model highlights the absence of the four helix bundle formed by Dsn1 and Nsl1. While the inhibitory N-terminal region of Dsn1 binding to Mis12, showed to prevent CENPC binding, is well predicted for *H. sapiens* and *S. cerevisiae*, it is not predicted and probably absent in the *A. thaliana* homolog.

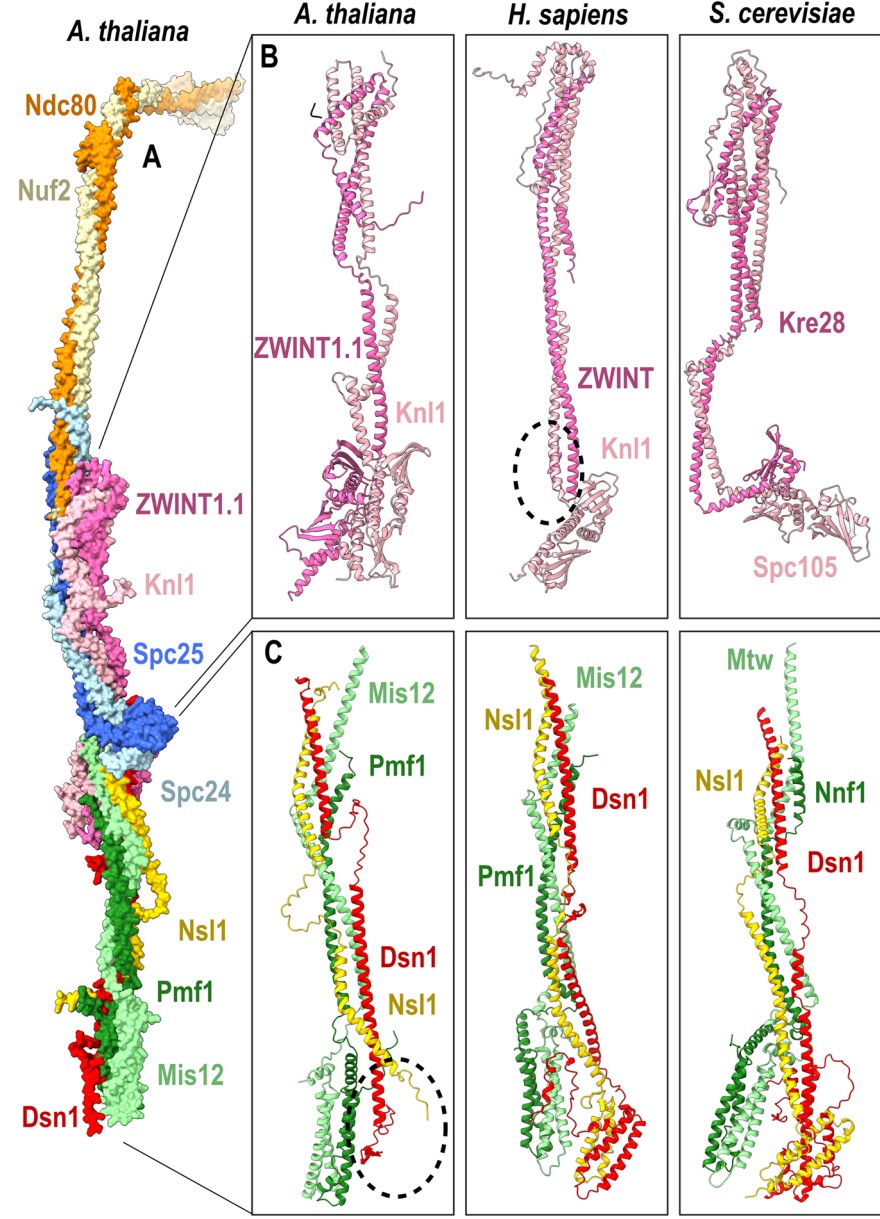

(region 15–74), predicted to bind the TPR domain of AtBMF1 with an ipTM of 0.65 (Fig. 7C). Remarkably, as shown in Fig. 6C, the region of CENPC predicted to interact with the MIS12 module (84–105) is located just downstream of the putative MAD3 interaction domain, making this structural arrangement possible and non-competitive. By contrast, in human (segment 1–32) and yeast (segment 1–41), the very N-terminal tail of CENPC interacts with the Mis12 module (Fig. 6C). This prediction of an interaction between the TPR domain of AtBMF1/MAD3 and AtCENPC was unexpected, since in human and yeast the TPR domain was shown instead to bind short disordered regions of KNL1. We did not detect any such interaction motifs between AtKNL1 and the TPR domain of AtBMF1. This may not be entirely surprising, as divergence in the way spindle assembly checkpoint (SAC) proteins such as BUB3 or BMF1 are recruited to the kinetochore had already been emphasized in the case of maize[38], where the region of interaction with BMF1 detected in maize Knl1 is conserved in monocots but shows high divergence in eudicots, suggesting distinct kinetochore architectures with SAC signaling in different plant lineages.

Finally, concerning CENPE, the vertebrate homolog interacts physically with both KNL1 and BubR1 (MAD3). However, no equivalent interaction could be detected for the *A. thaliana* proteins. In human, CENPE's N-terminus (1–1360) is linearly ubiquitinated and KNL1 (aa 300–338) binds those chains[62]. As this type of interaction is unlikely to be detected by AlphaFold3, it is difficult to state whether or not the CENPE-KNL1 might be conserved in Arabidopsis. By contrast, AlphaFold3 confidently predicts a direct interaction between the human MAD3 and CENPE proteins in regions of both proteins that are fully consistent with the interaction domains that were experimentally mapped in two previous studies[63,64]. Therefore, it is unlikely that we failed to identify an interaction with A. thaliana homologues due to a lack of sensitivity in AlphaFold3, suggesting that the MAD3-CENPE interaction is absent in plants. Human CENPE is more than twice as large as the *A. thaliana* homolog (2701 vs. 1273 residues) and the domain of CENPE predicted to interact with MAD3 in human is not conserved in *A. thaliana*. Instead, *A. thaliana* CENPE is predicted to adopt a relatively regular coiled-coil conformation throughout its sequence, unlike the human protein. The only significant interaction that we predicted is between a coiled-coil region of *A. thaliana* CENPE and the BUB3.3 WD40 β-propeller, with an ipTM score of 0.7 (Fig. 7D). Altogether, this suggests that the recruitment of atCENPE to the KMN complex is promoted by BUB3.3, which binds KNL1.

**Fig. 6 | Model of the MIS12-CSM1 interface.**
**A** Structural model obtained using AlphaFold3 of the complex between the MIS12 module and the Csm1 homodimer in A. thaliana mediated by the N-terminal tail of Dsn1 (red). **B** The structural model generated by AlphaFold3 for the homologous complex in S. cerevisiae exhibits a different binding mode with respect to *A. thaliana* consistent with the bound conformation observed in the experiment X-ray structure of a closely related complex Plowman et al.[20] of note, there is no detected homolog of Csm1 in *H. sapiens*. **C** Structural model generated by AlphaFold3 of the complex between the MIS12 module and the CENPC N-terminal disordered tail (brown color) in three species. In all cases, the region predicted with low PAE (higher confidence) corresponds to a short stretch of CENPC binding to the two-helix head of Mis12 subunit. The region of CENPC predicted for *H. sapiens* and *S. cerevisiae* is the same as the one identified experimentally and adopts a very similar conformation to that observed in the X-ray structure.

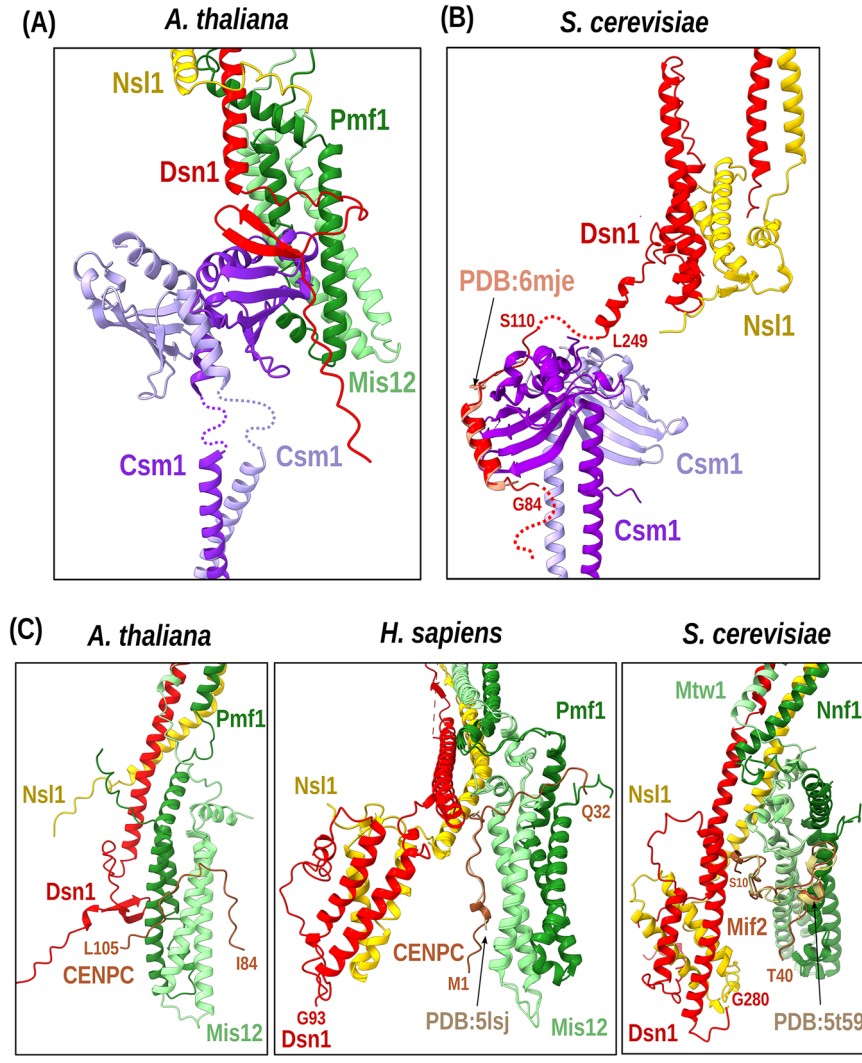

## Discussion

The KMN complex is a critical multi-subunit assembly in the outer kinetochore, essential for microtubule attachment and accurate chromosome segregation during mitosis and meiosis in eukaryotes. In mammals, this complex comprises ten subunits: MIS12, DSN1, NNF1, NSL1, NDC80, NUF2, SPC24, SPC25, KNL1, and ZWINT1. *Saccharomyces cerevisiae* has a homolog of all human subunits, and an additional subunit, CSM1. CSM1, together with DSN1, are crucial for the monopolar orientation of chromosomes at meiosis, in addition to their function in mitotic kinetochores. They directly interact with the monopolin complex in *Saccharomyces cerevisiae*, establish proper connection and cosegregation of sister kinetochores during meiosis I[20,21,65]. In plants several conserved subunits of KMN, including MIS12, NNF1, NDC80, NUF2, SPC24, SPC25[29,32–34], and recently KNL1 have been characterized[30,31]. However, subunits such as CSM1, DSN1, NSL1 and ZWINT1 which were suggested to be present in plants were yet to be functionally characterized[4,20,36].

Our study initially focused on three putative KMN subunits encoded by single-copy genes in Arabidopsis: DSN1, CSM1, and KNL1. In affinity purification coupled to mass spectrometry (AP-MS) experiments using DSN1, CSM1, and KNL1 as bait, we consistently identified all three, showing that they belong to the same complex in vivo. Additionally, all KMN complex subunits previously known in Arabidopsis (NDC80, NUF2, SPC24, SPC25, MIS12, and NNF1)[29,32–34] were reliably detected in these AP-MS experiments. Further, we showed that DSN1, CSM1, and KNL1 localize to the kinetochore, co-localizing with the centromere-specific marker CENH3. Finally, we showed that all three genes are essential, their mutation

leading to embryonic lethality, very similarly to the lethality reported for null mutant in *NDC80, NUF2, SPC24, SPC25, MIS12, and NNF1*[29,32–34,39,40] Together, this demonstrates that DSN1, CSM1, and KNL1 are essential members of the Arabidopsis KMN complex. Note that CSM1 was recently and independently characterized as a kinetochore protein[58].

In the same AP-MS experiments, with DSN1, CSM1, and KNL1 as bait, we identified two pairs of previously uncharacterized Arabidopsis KMN subunits, ZWINT1.1/.2 and NSL1.1/.2. ZWINT1 and NSL1 were previously suggested to be present in plants based on remote homology searches but were not experimentally confirmed[4,20,36].

ZWINT1.1 and ZWINT1.2 are closely related paralogs being located only 150 Kbp apart on chromosome 3 and encoding 421/428-amino acid proteins with 95% identity. cDNA sequencing showed that both are expressed, and specific peptides of each of them were identified in the AP-MS essays, showing that both ZWINT1.1 and ZWINT1.2 are associated with the kinetochore. *In planta* experiments showed that ZWINT1.1 forms foci at centromeres, colocalizing with CENH3, confirming that ZWINT1.1 is a kinetochore protein. Unique peptide counts for ZWINT1.1 were ~25% larger than for ZWINT1.2, suggesting that ZWINT1.1 is slightly more abundant at the kinetochore than ZWINT1.2. The analysis of *zwint1.1* and *zwint.2* mutants, including full gene deletions, showed that both *ZWINT1.1* and ZWINT1.2 are essential for plant development. Mutant embryos for *dsn1, csm1, knl1,* and other previously described KMN subunits die early in development leading to shrunken seeds. In contrast, *zwint1.1* and *zwint1.2* mutant embryos can develop, leading to wild-type-looking seeds that can germinate. However, *zwint1.1 or zwint1.2* resulting plantlets show a

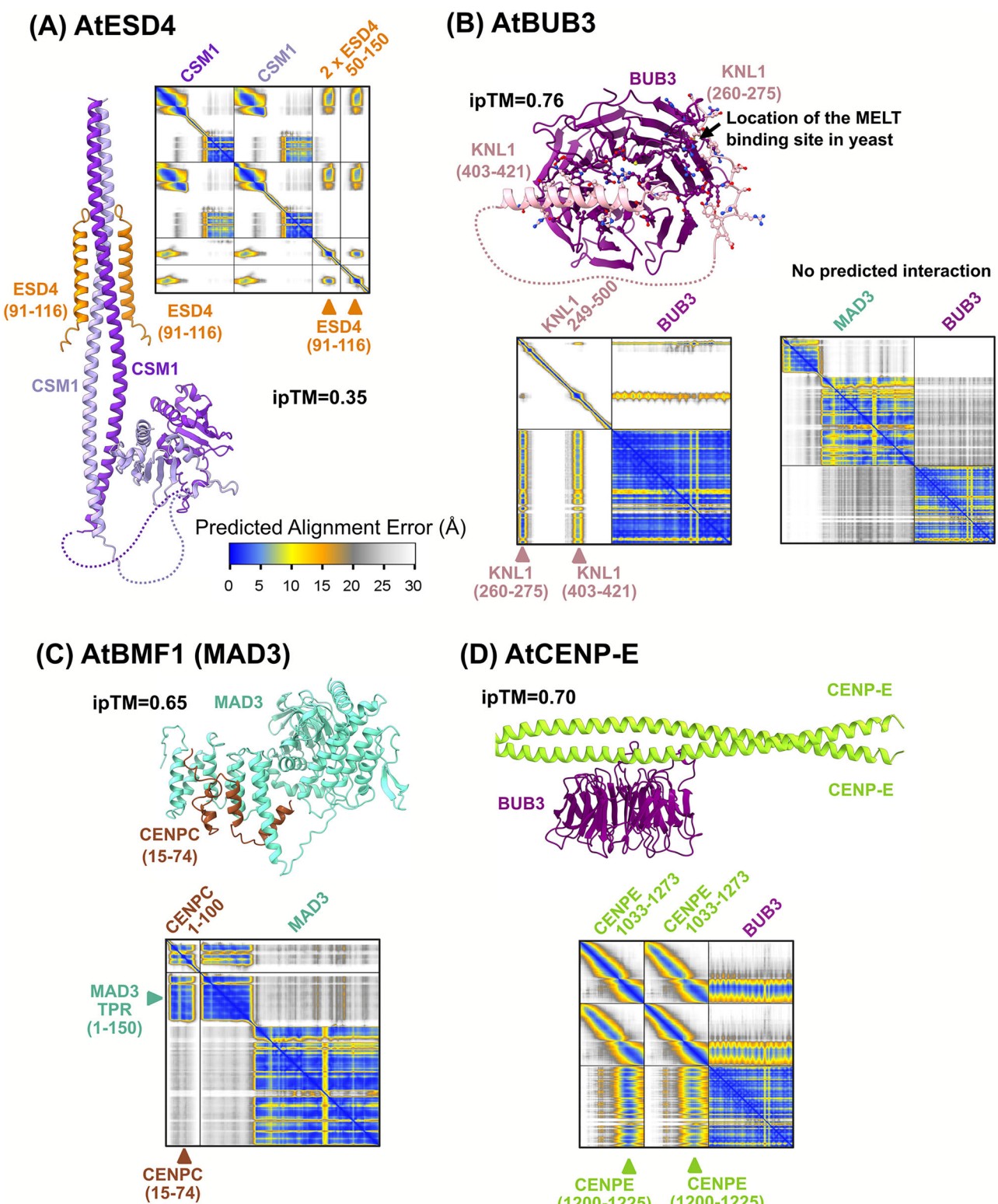

**Fig. 7 | Predicted complexes between four additional kinetochore subunits in *Arabidopsis thaliana*.** Each panel shows the predicted complex and its Predicted Alignment Error (PAE) heatmap which reports the confidence in the relative positions of residue pairs (blue = high to yellow = low confidence/flexible). Blue off-diagonal blocks indicate confident inter-domain positioning (i.e. a well-defined interface). **A** AtESD4 (orange)-AtCSM1 (lilac). A disordered region of ESD4 (residues 91–116) is suggested to mediate the interaction. The PAE indicates medium overall confidence (ipTM = 0.35, score often underestimated with coiled-coils). **B** AtBUB3 (purple)-AtKNL1 (pink). Two disordered KNL1 segments

(260–275 and 403–421) are predicted to contact BUB3 (ipTM = 0.76). The location of the MELT-binding site described in yeast is indicated for comparison. No interaction is predicted with AtBUB3/AtBMF1 (aquamarine). **C** AtBMF1/MAD3 (aquamarine)-AtCENP-C (brown). The TPR domain of MAD3 (residues 1–150) is predicted to bind a disordered N-terminal region of CENP-C (residues 15–74) (ipTM = 0.65) consistent with the PAE signal. **D** AtCENP-E (green)-BUB3 (purple). A coiled-coil region of CENP-E (residues 1200–1225) is predicted to interact with BUB3 (ipTM = 0.70) with a coherent high confidence in the PAE.

development arrest shortly after germination. We propose that the less severe lethality of *zwint1.1* and *zwint1.2* mutants compared with other KMN mutants is due to a partial redundancy between the two paralogues.

*NSL1.1* and *NSL1.2* encode proteins of 140 and 139 amino acids, with a significant divergence (43% sequence identity). Both NSL1.1 and NSL1.2 were detected in the AP-MS experiments; however, NSL1.1 was consistently identified in all triplicates of every five AP-MS with a total of 128 peptide counts, whereas NSL1.2 was detected only in a subset of the AP-MS and with a total peptide count of 22. This suggests that NSL1.1 is more abundant at kinetochore than NSL1.2. We showed that NSL1.1 colocalizes with the kinetochore marker CENH3 *in planta*, further supporting its identification as a kinetochore component[56]. NSL1 is essential for viability in distant eukaryotes including *Saccharomyces cerevisiae*, flies, and mammals[1,17,28,66,67]. Intriguingly, the single and double *nsl1 nsl2* Arabidopsis mutants we analyzed were viable, suggesting that NSL1 is not essential in Arabidopsis However, as the mutants used in our study are not complete gene deletions and have expression of truncated mRNA, we cannot rule out the possibility that some functionality is retained. Hypomorphic mutants, such as *knl1-2cr* and *knl1-3cr* in Arabidopsis, are viable but exhibit growth and developmental defects[30]. In our study, the CRISPR mutant *nsl1.1-1* provokes a frameshift after codon 37, leaving only ¼ of the N-ter protein potentially expressed (Supplementary Dataset 3). The next ATG codon (M) is the 99th codon, which if used as an alternative translation start would encode a C-ter protein of 41 amino acids. The N-ter 37 or C-Ter 41 amino acids in the *nsl1.1* mutant may be sufficient to support survival, but it seems unlikely. The *nsl1.2-1* allele results from a T-DNA insertion after the third codon, making it very unlikely that the N-ter of the encoded protein kept functionality. However, the 28th codon may serve as an alternative start codon, which would produce a truncated protein of 111 amino acids that may retain functionality. To rule out this possibility, and each a conclusion on the function of NSL1.1/NSL1.2, a complete deletion of both genes would be required, which was not achieved in the current study.

Our DSN1, CSM1, and KNL1 AP-MS experiments identified other interacting proteins, which are not predicted to be KMN subunits. This includes BUB3.3 and BMF1, which are involved in the spindle assembly checkpoint, and CENPC, which links the inner kinetochore to the KMN complex[23,42,43]. We also identified an Arabidopsis homolog of CENPE, despite its low sequence similarity to its human counterpart[48], It has been reported recently as KAK1, a Kinesin-7 protein, which plays a role in tubulin organization[44], suggesting a conserved role in microtubule capture and chromosome movement. Among the identified interactants was also ESD4, a SUMO protease, which suggests its possible role in kinetochore regulation through deSUMOylation. ESD4 was shown to localize mainly to the nuclear membrane and *esd4* mutants display severe growth defects (Figure S11)[52,53]. We also identified twenty-three other proteins (Table 1), which may represent novel kinetochore components or regulators, warranting further investigation. Among them, BIN4 belong to the topoisomerase VI complex, and, although distantly related to the human kinetochore protein CENP-U, has not been associated with kinetochore functions in plants[39,68,69]. Its co-precipitation with KNL1raises the possibility that BIN4 has retained some function at the kinetochore. One of the components of the KMN complex has been suggested to have a moonlighting role beyond its kinetochore function[29]. Thus, some of the proteins identified in the AP-MS experiments may be involved in potential non-kinetochore roles of KMN subunits.

Predictions of the structure of the Arabidopsis KMN complex show very similar global 3D organizations compared to yeast and humans. However, we noted that ZWINT1, DSN1 and NSL1 proteins show significant divergence with the loss and gain of specific domains. The comparative analysis of the models also reveals distinct interaction patterns involving the MIS12 module. In *A. thaliana*, the MIS12 module is predicted to form a complex with the CSM1 homodimer as in yeast, although their binding modes are predicted to have drastically diverged. In humans, CSM1 was not detected so far suggesting species-specific adaptations (Fig. 6). In contrast, the CENPC anchoring mechanism to the MIS12 module is conserved across species, highlighting a shared evolutionary feature of kinetochore assembly. In addition, we predicted interactions between the KMN complex and the associated proteins ESD4, BU3.3, BMF1 and CENPE, with substantial differences with the interactions described in humans. These models illustrate both conserved and divergent aspects of kinetochore architecture, reflecting evolutionary adaptations in different lineages. Future research could explore how these structural differences influence kinetochore assembly, function and regulation across eukaryotes.

Previous studies have shown that while some KMN subunits are highly conserved across kingdoms, others are challenging to identify due to low sequence similarity. By integrating interaction studies, remote homology detection, colocalization with centromeric proteins, mutant analysis, and Alpha Fold modeling, our findings provide evidence that the plant KMN complex closely mirrors the KMN complexes of *Saccharomyces cerevisiae* and mammals. One specific case is CSM1, which we confirmed to belong to the KMN complex in Arabidopsis as it does in *S. cerevisiae*, while it was not found in humans. This suggests that CSM1 was likely present in the common ancestor of eukaryotes[3] but was lost in the human lineage. Despite limited sequence similarity in some subunits, our results reveal a one-to-one correspondence of KMN subunits between *S. cerevisiae* and Arabidopsis. As the plant and fungi lineage diverged very early in eukaryote evolution, it suggests that the composition and structural organization of the KMN is inherited from the common ancestor of all living eukaryotes.

## Material and methods
### Genetic materials
The following Arabidopsis lines were used in this study: *titan9-2/csm1-1* (GK-374H01/N332674), *titan9-3/csm1-2* (SALKseq_77101/N882067), *titan9-4/csm1-3* (SALK_148785/N648785), *knl1-1* (SAIL_739_B08/ N833038), *knl1-2* (GK19_H10), *esd4* (SALK_032317/ N532317)[52,53], *nsl1.1-2* (salk_22495C/N656262), *nsl1.2-1* (GK_632E06), *zwint1.2-1* (Salk_138053/ N638053), *zwint1.2-2* (GK_291C04). The mutants *dsn1-1, dsn1-2, nsl1.1-1, zwint1.1-1, zwint1.1-2, zwint1.1-3, zwint1.2-3, zwint1.2-4, zwint1.2-5, and zwint1.2-6* were created in *Arabidopsis thaliana* Col-0 using the CRISPR-Cas9 system with guide RNAs designed to specifically modify the genomic sequences of their respective target genes (Supplementary Dataset 3 and 4).Two independent alleles of DSN1 possess mutations within its first exon: *dsn1-1* with a 194-base pair deletion, and *dsn1-2* with an adenine insertion at chr3: 10192895 position in Columbia-0 (TAIR10) genome, (Supplementary Dataset 3 and Fig. S1). The *nsl1.1-1* has a deletion of 112 bp from the first exon (Supplementary Dataset 3 and Fig. S1). All mutations result in a frameshift downstream of the open reading frame (ORF), encoding a truncated protein (Supplementary Dataset 3). The lengths of the deletions for *zwint1.1-1, zwint1.1-2, zwint1.1-3, zwint1.2-3, zwint1.2-4, zwint1.2-5, and zwint1.2-6* are as follows: 3106 bp, 2390 bp 2149 bp, 4807 bp, 4807 bp, 4807 bp and 4807 bp, respectively (Supplementary Dataset 3). The genotyping primers can be found in Supplementary Dataset 4.

### Statistics and reproducibility
Mendelian segregation was assesses using Chi-square tests. Filtering of the AP-MS results is described below.

### Cytological techniques
Siliques were fixed in 70% ethanol for at least one week, and seed counting was done manually on scanned images. To evaluate fertility, we counted the number of seeds per fruit on a minimum of five plants and ten fruits per plant.

Immunolocalization for Fig. 2 and 3 was carried out on somatic cells from floral buds with their three-dimensional structures preserved, as previously described[70,71]. The primary antibodies used were: Anti CSM1 raised in Guinea Pig against the peptide CHPRNYEDQSGKKQKR and affinity purified (Lab code PAK070; 1:200), anti-GFP raised in rabbit (cat# number ab6556, 1:250), anti-REC8 raised in rat (Lab code PAK0036; 1:250)[72], and anti-CENH3 raised in chicken against the peptide RTKHRVTRSQPRNQTDA and affinity purified (Lab code PAK008 1:500).

We used Abberior's secondary antibody, STAROrange (cat# STORANGE-1005) for anti-chicken antibodies and STARGreen for both anti-rabbit and anti-Guinea pig antibodies. GFP–CENH3 and CSM1- CENH3 co-foci images were taken with a Leica THUNDER Imager microscope and an Abberior STED microscope under a 100× oil immersion objective and then deconvolved and analyzed with the Huygens Essential version 20.04 (Scientific Volume Imaging, https://svi.nl/).

## CRISPR mutagenesis

Guide RNAs designed to specifically target the *DSN1*, *ZWINT1.1*, *ZWINT1.2* and *NSL1.1* genes were created using the TEFOR website at http://crispor.tefor.net. Two guide RNAs were used for *NSL1.1* as well as for *DSN1*, and four guide RNAs were employed for *ZWINT1.1* as well as for *ZWINT1.2* (Supplementary Date set 3 and 4). We used the Golden Gate two-level cloning method according to a reference[73]. Initially, we amplified the tracer RNA sequence from pICH86966 (adgene#48075) as a template. The forward primer consists of 23 bp specific to clone pICH86966, 20-base pair guide RNA recognition sequence, and BsaI site, accompanied by a specific overhang tailored (Supplementary Dataset 4). The reverse primer remained constant for all guide RNA cloning. In the first level of cloning, we utilized a BsaI-based Golden Gate assembly reaction, employing the AtU6 promoter from clone pICSL01009 (adgene #46968) and the amplified above-mentioned desired guide RNA into the chosen guide RNA vector. For dual guides, we selected positions 3 and 4 in vectors pICH47751 (48002) and pICH47761(adgene#48003). For quadruple guides, in addition to positions 3 and 4, we also used positions 5 in vector pICH47772 (adgene # 48004) and 6 in vector pICH47781 (adgene # 48005) for final destination vector PAGM4723(adgene # 48015). The complete assembly involved using Bpi-based Golden Gate, incorporating the FAST-Red clone (adgene # 117499, pICSL11015) as a seed coat selection marker positioned at 1, and the Cas9 clone (Adgene #153224, pAGM51323) featuring an enhanced SpCas9 under the AthRps5 promoter situated at 2. For dual guides, they occupied positions 3 and 4, while the quadruple guide RNA clone filled positions 3, 4, 5, and 6. The last position housed a specific linker clone: pICH41780 (adgene #48019) for two guides and pICH50927 (48049) for four guides.

## Protein tagging for localization

For the localization study, we isolated total RNA following the manufacturer's protocol, utilizing Trizol (Invitrogen). Subsequently, we conducted cDNA library synthesis using the Reverse Transcription System (Invitrogen SuperScript II) and oligo(dT) primers. The ORFs of DSN1, KNL1, ZWINT1.1, NSL1.1, and BUB3.3 were amplified from the cDNA library and subsequently inserted into the pGGC000 vector (Addgene #48858) as individual clones. We applied the green gate cloning method to insert all candidate genes with an N-terminal Venus tag under the ubiquitin promoter and ubiquitin terminator[74]. This assembly was achieved by utilizing six cloned subunits, which were combined in the ultimate destination vector, PAGM4723GG[75]. The subunits were arranged in the following order: Ubiquitin 10 promoter clone (pGGA006/Adgene# 48816), Venus clone in vector pGGB000 (Adgene# 136966), the desired kinetochore gene ORF clone in pGGC000 vector (Addgene #48858), a dummy vector clone (pGGD002/48834), and the Ubiquitin 10 terminator clone (pGGE009/Adgene#48841). Every clone, both for gene editing and Venus localization, underwent verification through Sanger sequencing to confirm the absence of mutations and to ensure an in-frame fusion.

## Proteins pull down

Arabidopsis cell suspension cultures expressing N-terminal GSrhino-tagged kinetochore protein bait and C-terminal GSrhino-tagged kinetochore protein were used for pull-down as previously described[76–78]. Experiments were performed in triplicate for each bait. Co-purified proteins were identified using standard protocols utilizing on bead-digested sample evaluated on a Q Exactive mass spectrometer (Thermo Fisher Scientific)[78]. The mass spectrometry proteomics data have been deposited to the ProteomeXchange Consortium via the PRIDE[79] partner repository with the dataset

identifier PXD062209 and 10.6019/PXD062209. After identification, quantitative analysis vs a large dataset of similar experiments with non-related baits[80] was performed, using the average normalized spectral abundance factors (NSAF). The Ln-transformed mean NSAF of all proteins identified in at least 2 replicates out of 3 were compared to the Ln-transformed mean NSAF of the same protein in the large dataset by a two-tailed t-test. Identifications were considered significantly enriched if they passed one of the following criteria: (i) two-peptide identifications present in at least two out of three replicates are significantly enriched if they were found with less than three other bait-groups in the large control AP-MS dataset or if they were enriched with a mean NSAF ratio $\geq 8$ AND a -Log10($p$ value) $\geq 8$, (ii) one-peptide identifications present in at least two out of three replicates, if they were significantly enriched with an Enrichment score (=mean NSAF ratio x -Log10($p$ value)) $\geq 1000$.

## Expression analysis

We used two independent plant seedlings for each genotype and their corresponding sister wild type, designated as biological replicates 1 and 2. Total RNA was isolated using the Machercy-Nagel NucleoSpin RNA Plant Mini Kit (Cat. No. - 740949) with DNase treatment, following the manufacturer's protocol. cDNA was synthesized using the Invitrogen SuperScript III Reverse Transcriptase (Cat. No. 18080044) with oligo(dT) primer. Semi-quantitative PCR was then performed, with GAP\DH serving as the internal normalization control. For mutant allele expression analysis, two sets of primers were used: one set flanking the mutation region and another set targeting a mutation-free region. The primers used were NSL1.1f1 and NSL1.1r1, NSL1.1f2 and NSL1.1r2, NSL1.2f1 and NSL1.2r1, NSL1.2f2 and NSL1.2r2, ZWINT1.2f1 and ZWINT1.2r1, ZWINT1.2f2 and ZWINT1.2r2, and GAPDH_F and GAPDH_R for GAPDH (Supplementary Dataset 4). PCR cycling conditions were as follows: initial denaturation at 95 °C for 2 min.; 35 cycles of 95 °C for 30 s, 55 °C for 30 s, and 72 °C for 25 s; final extension at 72 °C for 5 min. A 3% agarose gel was used to analyse the mutants and their corresponding sister wild-type samples for both replicates. For each RNA sample, a –RT control (reverse transcriptase omitted) was also included to check for genomic DNA contamination.

## Structural modeling

Structural models of the subunits isolated in the proteomic experiment were generated using both AlphaFold2 and AlphaFold3 programs and led to similar structures. Consequently, only the models obtained with Alpha-Fold3 are presented together with their validation scores (in Supplementary Dataset 5). Considering the *A. thaliana*, *H. sapiens* and *S. cerevisiae*, 13 models were generated in total. In addition to their scores, their delimitations of the subunits for every model are reported in Supplementary Dataset 5 Supplementary Fig. S12 also reports the Predicted Alignment Maps associated with each of these 13 models. The global model of the entire KMN was obtained from a combination of independent 3 models of the MIS12, KNL and NDC80 modules modules which we superimposed on the overlapping regions. This combination enabled to overcome a limit of the current AF3 program which bends erroneously large elongated structures, especially those made of coiled-coils. Figures (Figs. 5–7.) are represented using Chimera X software[81].

## Reporting summary

Further information on research design is available in the Nature Portfolio Reporting Summary linked to this article.

## Data availability

The IP-MS data are available in ProteomeXchange with identifier PXD062209 https://proteomecentral.proteomexchange.org/cgi/GetDataset?ID=PXD062209. All other data supporting the findings of this study are available within the paper and its Supplementary Information.

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

## Acknowledgements

We thank Dipti Vernekar for her suggestions on the manuscript and Dominique Eeckhoutt for data analysis. This work was supported by core funding from the Max Planck Society. D.K.S. is supported by the DBT-Ramalingaswami Re-entry Fellowship. D.K.S. also thanks to Director CSIR-CCMB Vinay K Nandicoori, for permitting access to the plant growth facility. This work was granted access to the HPC resources of IDRIS under the allocation 2024-AD010314343R1 to RG made by GENCI and to the BIOI2 platform resources at the I2BC.

## Author contributions

R.M. and D.K.S. led the project. B.W. and A.N. produced the genetic material. J.V.N. and G.D.J. performed and analysed the affinity purifications. S.D. and D.K.S performed and analysed the microscopy experiments. D.K.S. performed and analysed the genetic experiments. R.G. performed the structure predictions. D.K.S. and R.M. wrote the manuscript, incorporating contributions from all authors.

## Funding

## Competing interests

The authors declare no competing interests.
