## [Transparent Peer Review file · Communications Biology]

The composition and structure of the outer kinetochore KMN complex is conserved across Kingdoms

Corresponding Author: Professor Raphael Mercier

Version 0:

Reviewer comments:

Reviewer #1

(Remarks to the Author)

The composition and structure of the outer kinetochore KMN complex are conserved across Kingdoms. This study aims to define the composition of the outer kinetochore KMN complex in *Arabidopsis thaliana*, leveraging affinity purification-mass spectrometry (AP-MS), in planta protein localization, mutant analysis, and AlphaFold3-based structural modeling. The authors identify kinetochore components (AtDSN1, AtCSM1, AtNSL1A/B, and AtZWINT1A/B), propose conservation of KMN architecture across species, and provide evolutionary context. While the dataset is promising and of potential significance, the manuscript remains largely descriptive and lacks sufficient functional validation. Key predicted interactions remain unvalidated experimentally, and mutant phenotypic analyses are limited. Incorporating experimental interaction studies, genetic rescue experiments, and cytological evidence supporting the functional role of the proposed KMN subunits would greatly strengthen the study's mechanistic insight and overall impact.

Below are suggestions for improving the manuscript:

1. Page 4, Line 92-95: "We performed five independent AP-MS experiments in triplicate, covering N- and C-terminal GSRhino fusions of CSM1 and DSN1, whereas for KNL1 the N-terminal fusion was analyzed".

Reviewer: The manuscript uses both N- and C-terminal GSRhino tags for two proteins but only the N-terminal tag for KNL1; clarification on this choice would be useful.

2. Page 4, Line 101-103: "Strikingly, 19 proteins were robustly identified in at least three out of five experiments, with 16 of them being systematically identified in all five AP-MS experiments."

Reviewer: It would be helpful to clarify whether the analysis aimed specifically to identify common kinetochore proteins across all AP-MS experiments or to profile the broader protein interactome. Please provide a rationale for the selected experimental conditions, particularly if some baits are known to interact with a wide range of partners. Additionally, I suggest conducting Gene Ontology and KEGG pathway analyses for each AP-MS dataset to enable more detailed functional insights into the pull-down proteins. Including a protein-protein interaction network map would also enhance the visualization and interpretation of the identified complexes.

3. Page 6, Line 168-172: "we tagged DSN1, KNL1, NSL1A, ZWINT1A, BUB3.3, and GUS as negative control with the fluorescent protein Venus (variant of the Green Fluorescent Protein GFP) 53 at the N-terminus under the ubiquitin10 promoter and transformed plants. We used three transformed plants per tagged protein for localization analysis."

Reviewer: Please clarify why the localization of ZWINT1B and NSL1B was not tested. Was this decision based on their sequence similarity to ZWINT1A and NSL1A, their lower abundance, or other technical or biological factors? In the preprint manuscript deposited on BioRxiv (see link below), NSL1 and ZWINT1 were defined as NSL1.1/NSL1.2 and ZWINT1.1/ZWINT1.2, respectively.

To ensure consistency and avoid confusion in future references, we kindly request that you adopt the same nomenclature. Furthermore, the study linked below provides insights into the subcellular localization of these proteins throughout the mitotic cell cycle. We encourage you to refer to this work: <https://www.biorxiv.org/content/10.1101/2024.11.04.621965v1>

4. Page 6, Line 172-175: "Immunolocalization was performed with anti-GFP antibodies on fixed tissue from unopened floral buds to detect the tagged proteins. CSM1 localization was studied using an anti-CSM1 polyclonal antibody raised against a

specific peptide."

Reviewer: The BioRxiv manuscript describes the dynamic behavior of DSN1, KNL1, NSL1A, and ZWINT1A during the cell cycle, but does not include corresponding data for CSM1 (<https://www.biorxiv.org/content/10.1101/2024.11.04.621965v1>). We encourage you to consider analyzing the localization of CSM1 during mitosis to determine whether it associates with centromeres throughout the cell cycle, rather than only in interphase. For other KMN components, please refer to the BioRxiv preprint.

Additionally, please clarify why CSM1 localization was assessed using antibodies, while a GFP fusion approach was employed for the other proteins. Consistency in methodology would facilitate more direct comparisons.

5. Page 7, Line 197: "Most KMN genes are essential". While the genetic analysis provides strong initial evidence for the essential roles of CSM1, DSN1, KNL1, ZWINT1A/B, NSL1A/B in Arabidopsis, several technical and biological aspects remain underexplored."

Reviewer: First, there is no confirmation of transcript or protein loss or reduction in the CRISPR or T-DNA insertion mutants; validating the knockout status using RT-PCR or Western blot would strengthen the conclusions.

6. Page 7, Line 201-211: "However, their fruits contained approximately 25% shrunken seeds (Figure 2 and source data set 2). This observation is consistent with the expected Mendelian segregation ratio of 3:1 (Chi-square tests, p-values of 0.80 for *knl1*, 0.30 for *csm1*, and 0.86 for *dsn1*), indicating that the quarter of homozygous embryos are not viable."

Reviewer: For the embryo lethal mutants such as *csm1*, *dsn1*, and *knl1*, microscopic analysis of embryo development stages can be performed. DIC or confocal imaging could help pinpoint the timing of developmental arrest.

7. Page 8, Line 236-239: "When the seeds from heterozygous plants were sown in vitro, we observed plantlets dying shortly after germination for the three *zwint1a* alleles (Figure S6), with a frequency compatible with 1:4 (dead/viable plants: *zwint1a-1* 17/79, *zwint1a-2* 21/78, *zwint1a-3* 20/76)."

Reviewer: Please genotype *zwint1a* seedlings shortly after germination, prior to lethality to confirm their homozygosity. This would allow early-stage functional analyses, such as assessing mitotic defects or kinetochore protein localization (e.g., CENH3) by immunostaining.

8. Page 9, Line 258-266: "We then combined *nsl1a-1* and *nsl1b-1* mutations, which both affect the first exon of their respective gene. In the F2 progeny of double heterozygous *nsl1a-1* -/+ *nsl1b-1* -/+, we obtained 4 double mutants among 96 plants, which was close to the expected number under Mendelian segregation (6 expected). The double mutant is viable and does not show major defects, though it showed a slower growth (Figure S7) and 8.8% seed abortion in fruits (Figure 2). However, as *nsl1a-1* and *nsl1b-1* are not full deletions, we cannot exclude that some functionality was retained in the mutant."

Reviewer: For NSL1A and NSL1B, the alleles used may not represent complete nulls, and residual gene expression could explain the mild phenotype; transcript or protein quantification is needed to rule this out. No cytological analyses have been performed in the double mutants exhibiting delayed growth. Assessing root lengths, mitotic defects, and localization of key subunits (e.g., CENH3, NDC80) in the mutants would clarify whether the phenotypes reflect defects in kinetochore assembly or function. These analyses would provide important insight into functional dependencies within the complex and enhance the biological impact of the study.

9. Page 9-10, Line 280-283, 301-303: "In *A. thaliana*, the first RWD domain in the tandem is predicted to interact with the C-terminal helix of Mis12, whereas in *H. sapiens*, the absence of the RWD tandem in ZWINT is associated with a shorter Mis12 homolog. AF3 (Figure 4) predicts AtCSM1 binds to KMN through the N-terminal extension of Dsn1, similar to *S. cerevisiae*. However, in *A. thaliana*, DSN1 is predicted to fold as a hairpin upon CSM1 binding."

Reviewer: The structural predictions generated using AlphaFold3 offer valuable insights into the organization and evolution of the KMN complex in *Arabidopsis thaliana*, yet they remain purely computational and lack experimental validation. Key predicted interactions, such as those between CSM1 and DSN1 or ZWINT1 and MIS12, are not supported by at least one of the biochemical or interaction assays (e.g., co-immunoprecipitation or Yeast-two hybrid), limiting the confidence in these models.

10. Page 13, Line 400-406: "Predictions of the structure of the Arabidopsis KMN complex show very similar global 3D organizations compared to yeast and humans. However, we noted that ZWINT1, DSN1, and NSL1 proteins show significant divergence with the loss and gain of specific domains. The comparative analysis of the models also reveals distinct interaction patterns involving the MIS12 module. In *A. thaliana*, the MIS12 module is predicted to form a complex with the CSM1 homodimer as in yeast, although their binding modes are predicted to have drastically diverged."

Reviewer: The study highlights several structural divergences compared to yeast and human complexes, such as loss and gain of specific domains in KMN subunits. There is minimal discussion of how these differences may impact kinetochore assembly or regulatory mechanisms in plants.

11. The discussion of interaction analysis and mutant phenotypes could be further strengthened by a more extensive comparison with previously characterized kinetochore components. Several mutants of essential kinetochore subunits (e.g.,

NDC80, MIS12, NNF1, SPC24/25) display phenotypes similar to those reported for CSM1, DSN1, and KNL1. Integrating these comparisons would not only contextualize the findings but also provide broader insights into the functional architecture and evolutionary conservation of kinetochore mechanisms in plants.

Minor Issues

- Use standard formatting (e.g., Table 1 instead of Table 1).
- Line 130, 133: "ZWINTA and five specific to ZWINT1B.", "ZWINTA is slightly more abundant than ZWINT1B" → should be: "ZWINT1A" (ensure all protein names are consistent)
- Several errors such as species name and protein names in Table1 and 2 (eg., Arabidopsis instead of arabidopsis or DSN1 instead of DNS1)
- Statistical analysis is missing in the methods section. Additionally, including more subheadings would improve readability and help organize the methods section more clearly.

Reviewer #2

(Remarks to the Author)

This study provides a welcome, complete view of the KMN complex in Arabidopsis. Although the results were largely anticipated by other studies, this work is notably comprehensive, the experiments are well executed, and the paper is well written. The work includes well-controlled mass spec, detailed cellular localization, extensive mutant analysis (including new CRISPR alleles), and AlphaFold analyses of the entire complex that elevate the quality of the paper. The plant kinetochore field has been circling around this type of analysis for years; it is nice to see it finally done and put together so nicely. The paper is likely to be highly cited.

I enjoyed the paper and have little to offer in terms of improvement. One thing that seems missing is that when they get to the AlphaFold section, four interesting non-core KMN proteins, BUB3, BMF1, CENP-E and ESD4 are not included. It would add completeness to say whether they tried to identify binding partners for these three proteins in the general KMN model. If they tried and found nothing convincing, that is fine, but it would be good to say so.

Reviewer #3

(Remarks to the Author)

This manuscript presents a comprehensive characterization of the outer kinetochore components in Arabidopsis thaliana, with a specific focus on identifying all members of the KMN complex using affinity purification. The authors have successfully identified six previously uncharacterized KMN components and complemented their biochemical findings with localization and functional analyses. Overall, this is an excellent and timely study that significantly advances our understanding of outer kinetochore architecture in plants and draws informative parallels with corresponding structures in yeast and animals.

Major Comments

1. Additional Localization Analysis

The authors have provided localization data for interphase cells and shown that AtDsn1, AtCsm1, AtNSL1A, and AtZWINT1A co-localize with cenH3, a centromere-specific histone variant. However, to conclusively demonstrate that these proteins are bona fide kinetochore components, I recommend including additional localization data from mitotic cells. Time-lapse live-cell imaging or immunolocalization in fixed cells would strengthen the evidence for kinetochore association of these newly identified KMN components during active cell division.

2. Additional Functional Characterization of AtZWINT1A

The functional analyses using T-DNA insertion and CRISPR-generated mutants are commendable, and the conclusion that most genes are essential is well-supported. The phenotype of AtZWINT1A mutants, which exhibit post-germination lethality, is particularly intriguing. I suggest including a simple cytological analysis, such as DAPI staining of root meristem cells, to visualize mitotic cells. This would help determine whether phenotype is associated with chromosome missegregation or other mitotic defects and would provide direct evidence into the kinetochore role of AtZWINT1A.

Minor Comments

3. Abstract Wording

The sentence: "The outer kinetochore attaches to the microtubules and is named after three components KNL1C, MIS12C, and NDC80C (KMN)."

could be revised for clarity. I recommend replacing "components" with either "sub-complexes" or "subunits," as each of KNL1C, MIS12C, and NDC80C represents a multi-protein assembly.

4. Introduction Improvements

The introduction would benefit from a more detailed overview of the plant kinetochore structure. Including recent review articles (Xie Y, Wang M, Mo B, Liang C. Plant kinetochore complex: composition, function, and regulation. Front Plant Sci. 2024 Oct 10;15:1467236. doi: 10.3389/fpls.2024.1467236 ; Kozgunova E. Recent advances in plant kinetochore research. Front Cell Dev Biol. 2025 Jan 22;12:1510019. doi: 10.3389/fcell.2024.1510019) specific to plant kinetochore organization would enhance the context and strengthen the rationale for the study.

5. BIN4 and Non-Canonical Interactions of KMN complex

The identification of proteins not directly implicated in cell division, particularly BIN4, is an interesting finding. BIN4, although distantly related to CENP-U, has not been associated with centromere or kinetochore functions in plants. Its co-precipitation with KMN components raises intriguing questions about potential indirect or novel interactions. I encourage the authors to mention this explicitly in the manuscript and offer possible interpretations, even if speculative, as this could inspire future functional investigations.

Previous studies suggest that certain components of the Mis12 complex, such as Nnf1 (Allipra, S., Anirudhan, K., Shivanandan, S., Raghunathan, A. and Maruthachalam, R. (2022), The kinetochore protein NNF1 has a moonlighting role in the vegetative development of *Arabidopsis thaliana*. *Plant J*, 109: 1064-1085. <https://doi.org/10.1111/tpj.15614>), may play roles beyond their canonical kinetochore functions. In light of this, I am curious whether any of the proteins identified through the pull-down assay—particularly those not directly involved in cell division—might provide insights into potential non-kinetochore roles of the KMN complex or its interacting partners. Exploring or at least acknowledging this possibility could enrich the discussion and broaden the implications of the findings.

Version 1:

Reviewer comments:

Reviewer #1

(Remarks to the Author)

Dear Editor,

I have carefully reviewed the revised version of the manuscript. The authors have addressed all my previous comments satisfactorily, and I have no further remarks. I consider the manuscript suitable for publication in its current form.

Best regards,
Inna Lermontova

Reviewer #3

(Remarks to the Author)

The authors have sufficiently addressed the concerns raised in my previous review. They have incorporated additional localization data, clarified methodological choices, and expanded on the functional analysis of various KMN complex components in *Arabidopsis thaliana*. The revisions provide a more comprehensive view of the KMN complex, which significantly strengthens the manuscript.

Overall, the manuscript has been significantly improved in terms of content, clarity, and detail. I have no further major concerns and believe the manuscript is now ready for publication.

Reviewers' comments:

Reviewer #1 (Remarks to the Author):

The composition and structure of the outer kinetochore KMN complex are conserved across Kingdoms

This study aims to define the composition of the outer kinetochore KMN complex in *Arabidopsis thaliana*, leveraging affinity purification-mass spectrometry (AP-MS), in planta protein localization, mutant analysis, and AlphaFold3-based structural modeling. The authors identify kinetochore components (AtDSN1, AtCSM1, AtNSL1A/B, and AtZWINT1A/B), propose conservation of KMN architecture across species, and provide evolutionary context. While the dataset is promising and of potential significance, the manuscript remains largely descriptive and lacks sufficient functional validation. Key predicted interactions remain unvalidated experimentally, and mutant phenotypic analyses are limited. Incorporating experimental interaction studies, genetic rescue experiments, and cytological evidence supporting the functional role of the proposed KMN subunits would greatly strengthen the study's mechanistic insight and overall impact.

> Thank you for your thorough evaluation of the work and for your insightful suggestions. While we agree that this manuscript opens doors for further research, we also believe that the results presented here are significant enough to be shared with the community, where they are likely to trigger additional projects.

Below are suggestions for improving the manuscript:

1. Page 4, Line 92-95: "We performed five independent AP-MS experiments in triplicate, covering N- and C-terminal GSRhino fusions of CSM1 and DSN1, whereas for KNL1 the N-terminal fusion was analyzed".

Reviewer: The manuscript uses both N- and C-terminal GSRhino tags for two proteins but only the N-terminal tag for KNL1; clarification on this choice would be useful.

> There was no specific reason for this choice beyond the optimization of resources. The results that very similar results were obtained with N- and C-terminal GSRhino fusions for CSM1 and DSN1 did not encourage us to perform both for KNL1.

2. Page 4, Line 101-103: "Strikingly, 19 proteins were robustly identified in at least three out of five experiments, with 16 of them being systematically identified in all five AP-MS experiments."

Reviewer: It would be helpful to clarify whether the analysis aimed specifically to identify common kinetochore proteins across all AP-MS experiments or to profile the broader protein interactome. Please provide a rationale for the selected experimental conditions, particularly if some baits are known to interact with a wide range of partners. Additionally, I suggest conducting Gene Ontology and KEGG pathway

analyses for each AP-MS dataset to enable more detailed functional insights into the pull-down proteins. Including a protein–protein interaction network map would also enhance the visualization and interpretation of the identified complexes.

> We did not aim to only identify kinetochore proteins. Table1 contains all the identified proteins (after unbiased filtering). We clarified this point in the revised manuscript. The very strong enrichment in kinetochore proteins is an experimental result. As suggested, we have included a protein–protein interaction network map in the revised manuscript.

3. Page 6, Line 168-172: “we tagged DSN1, KNL1, NSL1A, ZWINT1A, BUB3.3, and GUS as negative control with the fluorescent protein Venus (variant of the Green Fluorescent Protein GFP) 53 at the N-terminus under the ubiquitin10 promoter and transformed plants. We used three transformed plants per tagged protein for localization analysis.”

Reviewer: Please clarify why the localization of ZWINT1B and NSL1B was not tested. Was this decision based on their sequence similarity to ZWINT1A and NSL1A, their lower abundance, or other technical or biological factors?

This decision was indeed based on sequence similarity. The rationale was that demonstrating localization to the kinetochore of one of the two paralogs is sufficient to support a kinetochore function. Investigating the localization of the others would be interesting but is beyond the scope of this study.

In the preprint manuscript deposited on BioRxiv (see link below), NSL1 and ZWINT1 were defined as NSL1.1/NSL1.2 and ZWINT1.1/ZWINT1.2, respectively.

To ensure consistency and avoid confusion in future references, we kindly request that you adopt the same nomenclature.

> This is a good point. We renamed genes/proteins to NSL1.1/NSL1.2 and ZWINT1.1/ZWINT1.2 in the revised manuscript.

Furthermore, the study linked below provides insights into the subcellular localization of these proteins throughout the mitotic cell cycle. We encourage you to refer to this work: <https://www.biorxiv.org/content/10.1101/2024.11.04.621965v1>

> Thank you for pointing this. We refer to this preprint in the revised manuscript.

4. Page 6, Line 172-175: “Immunolocalization was performed with anti-GFP antibodies on fixed tissue from unopened floral buds to detect the tagged proteins. CSM1 localization was studied using an anti-CSM1 polyclonal antibody raised against a specific peptide.”

Reviewer: The BioRxiv manuscript describes the dynamic behavior of DSN1, KNL1,

NSL1A, and ZWINT1A during the cell cycle, but does not include corresponding data for CSM1 (<https://www.biorxiv.org/content/10.1101/2024.11.04.621965v1>).

We encourage you to consider analyzing the localization of CSM1 during mitosis to determine whether it associates with centromeres throughout the cell cycle, rather than only in interphase. For other KMN components, please refer to the BioRxiv preprint.

> We have now included localization of CSM1 at both mitosis and meiosis, showing that it associates to the kinetochore throughout the cell cycle. We further show using super-resolution that CSM1 localizes to the outer kinetochore at metaphase (Figure 3). We also refer to Pettkó-Szandtner et al in the revised manuscript.

Additionally, please clarify why CSM1 localization was assessed using antibodies, while a GFP fusion approach was employed for the other proteins. Consistency in methodology would facilitate more direct comparisons.

> We developed an antibody against CSM1 to facilitate co-detection with other proteins in future studies. We believe that the conclusion that proteins localize to kinetochores can be equally assessed using GFP fusions and immunolabeling.

5. Page 7, Line 197: "Most KMN genes are essential". While the genetic analysis provides strong initial evidence for the essential roles of CSM1, DSN1, KNL1, ZWINT1A/B, NSL1A/B in Arabidopsis, several technical and biological aspects remain underexplored."

Reviewer: First, there is no confirmation of transcript or protein loss or reduction in the CRISPR or T-DNA insertion mutants; validating the knockout status using RT-PCR or Western blot would strengthen the conclusions.

> These are valid points. We now provide RT-PCR data for *zwint1.2*, *ns1.1* and *ns1.2* mutants (Figure S7-S9).

However, we cannot use Western blot because we don't have an antibody for most of them, and the *CSM1* mutation is lethal. RT-PCR is also not amenable to non-viable mutants. Our *ZWINT1A* reference mutant has a full genomic deletion, which makes RT-PCR irrelevant. However,

In addition, we generated a novel series of full deletion allele of *zwint1.2*, which leads to lethality, suggesting that *ZWINT1.2* is also an essential gene and that the two described TDNA alleles are not null.

6. Page 7, Line 201-211: "However, their fruits contained approximately 25% shrunken seeds (Figure 2 and source data set 2). This observation is consistent with the expected Mendelian segregation ratio of 3:1 (Chi-square tests, p-values of 0.80 for *kn1*, 0.30 for *csm1*, and 0.86 for *dsn1*), indicating that the quarter of homozygous

embryos are not viable.”

Reviewer: For the embryo lethal mutants such as *csn1*, *dsn1*, and *kn1*, microscopic analysis of embryo development stages can be performed. DIC or confocal imaging could help pinpoint the timing of developmental arrest.

> This is an interesting question, but we consider it to be outside the scope of the present study.

7. Page 8, Line 236-239: “When the seeds from heterozygous plants were sown in vitro, we observed plantlets dying shortly after germination for the three *zwint1a* alleles (Figure S6), with a frequency compatible with 1:4 (dead/viable plants: *zwint1a-1* 17/79, *zwint1a-2* 21/78, *zwint1a-3* 20/76).”

Reviewer: Please genotype *zwint1a* seedlings shortly after germination, prior to lethality to confirm their homozygosity. This would allow early-stage functional analyses, such as assessing mitotic defects or kinetochore protein localization (e.g., CENH3) by immunostaining.

> We genotyped *zwint1a* seedlings shortly after germination and confirmed that the non-developing plantlets are all homozygous for the mutation (n=15). This is included in the revised manuscript.

8. Page 9, Line 258-266: “We then combined *ns1a-1* and *ns1b-1* mutations, which both affect the first exon of their respective gene. In the F2 progeny of double heterozygous *ns1a-1* -/+ *ns1b-1* -/+, we obtained 4 double mutants among 96 plants, which was close to the expected number under Mendelian segregation (6 expected). The double mutant is viable and does not show major defects, though it showed a slower growth (Figure S7) and 8.8% seed abortion in fruits (Figure 2). However, as *ns1a-1* and *ns1b-1* are not full deletions, we cannot exclude that some functionality was retained in the mutant.”

Reviewer: For NSL1A and NSL1B, the alleles used may not represent complete nulls, and residual gene expression could explain the mild phenotype; transcript or protein quantification is needed to rule this out.

> We fully agree and pointed this limitation in the original manuscript. However, we have even more clearly stated this limitation in our revised manuscript. Indeed, some mRNA expression is detected in all *NSL1* mutants. Unfortunately, neither transcription nor protein quantification would allow us to rule out residual functionality, as it is never possible to exclude the presence of undetected transcripts or proteins. Only full deletions would allow reaching this conclusion, as we mention in the revised manuscript. We are aiming to produce these full deletions but have unfortunately not succeeded so far; achieving this would require more time than this revision allows and this would need to be the focus of futures studies.

No cytological analyses have been performed in the double mutants exhibiting delayed growth. Assessing root lengths, mitotic defects, and localization of key subunits (e.g., CENH3, NDC80) in the mutants would clarify whether the phenotypes reflect defects in kinetochore assembly or function. These analyses would provide important insight into functional dependencies within the complex and enhance the biological impact of the study.

> The growth defect is relatively subtle, suggesting that the localization of kinetochore proteins is only slightly affected, which makes it challenging to diagnose. We believe the priority here is developing additional alleles.

9. Page 9-10, Line 280-283, 301-303: "In *A. thaliana*, the first RWD domain in the tandem is predicted to interact with the C-terminal helix of Mis12, whereas in *H. sapiens*, the absence of the RWD tandem in ZWINT is associated with a shorter Mis12 homolog. AF3 (Figure 4) predicts AtCSM1 binds to KMN through the N-terminal extension of Dsn1, similar to *S. cerevisiae*. However, in *A. thaliana*, DSN1 is predicted to fold as a hairpin upon CSM1 binding."

Reviewer: The structural predictions generated using AlphaFold3 offer valuable insights into the organization and evolution of the KMN complex in *Arabidopsis thaliana*, yet they remain purely computational and lack experimental validation. Key predicted interactions, such as those between CSM1 and DSN1 or ZWINT1 and MIS12, are not supported by at least one of the biochemical or interaction assays (e.g., co-immunoprecipitation or Yeast-two hybrid), limiting the confidence in these models.

> The pull-downs confirm that these proteins interact *in vivo*, though they cannot distinguish between direct and indirect interactions. AlphaFold3 predictions complement this nicely, even though they are only computational. Exploring these interactions in detail will be the topic of future studies. Besides, the predicted CSM1-DSN1 interaction was very recently confirmed using Yeast-two hybrid in an independent study focusing on CSM1 (Zhang et al, *iScience* ; 08/2025). We cite this study in the revised manuscript.

10. Page 13, Line 400-406: "Predictions of the structure of the *Arabidopsis* KMN complex show very similar global 3D organizations compared to yeast and humans. However, we noted that ZWINT1, DSN1, and NSL1 proteins show significant divergence with the loss and gain of specific domains. The comparative analysis of the models also reveals distinct interaction patterns involving the MIS12 module. In *A. thaliana*, the MIS12 module is predicted to form a complex with the CSM1 homodimer as in yeast, although their binding modes are predicted to have drastically diverged."

Reviewer: The study highlights several structural divergences compared to yeast and human complexes, such as loss and gain of specific domains in KMN subunits. There

is minimal discussion of how these differences may impact kinetochore assembly or regulatory mechanisms in plants.

11. The discussion of interaction analysis and mutant phenotypes could be further strengthened by a more extensive comparison with previously characterized kinetochore components. Several mutants of essential kinetochore subunits (e.g., NDC80, MIS12, NNF1, SPC24/25) display phenotypes similar to those reported for CSM1, DSN1, and KNL1. Integrating these comparisons would not only contextualize the findings but also provide broader insights into the functional architecture and evolutionary conservation of kinetochore mechanisms in plants.

> This is included in the revised manuscript

Minor Issues

- Use standard formatting (e.g., Table 1 instead of Table 1).
- Line 130, 133: “ZWINTA and five specific to ZWINT1B.”, “ZWINTA is slightly more abundant than ZWINT1B” → should be: “ZWINT1A” (ensure all protein names are consistent)
- Several errors such as species name and protein names in Table 1 and 2 (eg., Arabidopsis instead of arabidopsis or DSN1 instead of DNS1)
- Statistical analysis is missing in the methods section. Additionally, including more subheadings would improve readability and help organize the methods section more clearly.

> Corrected

Reviewer #2 (Remarks to the Author):

This study provides a welcome, complete view of the KMN complex in Arabidopsis. Although the results were largely anticipated by other studies, this work is notably comprehensive, the experiments are well executed, and the paper is well written. The work includes well-controlled mass spec, detailed cellular localization, extensive mutant analysis (including new CRISPR alleles), and AlphaFold analyses of the entire complex that elevate the quality of the paper. The plant kinetochore field has been circling around this type of analysis for years; it is nice to see it finally done and put together so nicely. The paper is likely to be highly cited.

> Thank you for your positive assessment of the work.

I enjoyed the paper and have little to offer in terms of improvement. One thing that seems missing is that when they get to the AlphaFold section, four interesting non-core KMN proteins, BUB3, BMF1, CENP-E and ESD4 are not included. It would add completeness to say whether they tried to identify binding partners for these three proteins in the general KMN model. If they tried and found nothing convincing, that is fine, but it would be good to say so.

> Thank you for the suggestion; we have included this analysis in the revised manuscript (see Figure 7 and the associated text). Briefly, we predicted interactions between ESD4 and CSM1, BU3.3 and KNL1, BMF1 and CENPC, and CENPE and BUB3.3. Please refer to the manuscript for the full analysis.

Reviewer #3 (Remarks to the Author):

This manuscript presents a comprehensive characterization of the outer kinetochore components in *Arabidopsis thaliana*, with a specific focus on identifying all members of the KMN complex using affinity purification. The authors have successfully identified six previously uncharacterized KMN components and complemented their biochemical findings with localization and functional analyses. Overall, this is an excellent and timely study that significantly advances our understanding of outer kinetochore architecture in plants and draws informative parallels with corresponding structures in yeast and animals.

>Thank you for your positive assessment of the work

Major Comments

1. Additional Localization Analysis

The authors have provided localization data for interphase cells and shown that AtDsn1, AtCsm1, AtNSL1A, and AtZWINT1A co-localize with cenH3, a centromere-specific histone variant. However, to conclusively demonstrate that these proteins are bona fide kinetochore components, I recommend including additional localization data from mitotic cells. Time-lapse live-cell imaging or immunolocalization in fixed cells would strengthen the evidence for kinetochore association of these newly identified KMN components during active cell division.

> Thank you for the suggestion. In the revised manuscript, we have added immunolocalization of CSM1 in both mitosis and meiosis, showing that it associates with the kinetochore throughout the cell cycle. Note that a work very recently published independently showed that CSM1 is constitutively associated with the kinetochore (Zhang et al, Iscience 2025). We further showed, using super-resolution, that CSM1 localizes to the outer side of the kinetochore (compared to CENH3. Figure 3).

As reviewer #1 suggested, we refer to Pettkó-Szandtner et al. (<https://www.biorxiv.org/content/10.1101/2024.11.04.621965v1>) for the localization of DSN1, KNL1, NSL1A, and ZWINT1A throughout the cell cycle. This confirms that they are constitutively associated with the kinetochore.

2. Additional Functional Characterization of AtZWINT1A

The functional analyses using T-DNA insertion and CRISPR-generated mutants are commendable, and the conclusion that most genes are essential is well-supported. The phenotype of AtZWINT1A mutants, which exhibit post-germination lethality, is particularly intriguing. I suggest including a simple cytological analysis, such as DAPI staining of root meristem cells, to visualize mitotic cells. This would help determine whether phenotype is associated with chromosome missegregation or other mitotic defects and would provide direct evidence into the kinetochore role of AtZWINT1A.

> We have tried to perform this analysis but unfortunately failed to produce relevant images of mitotic cells.

Minor Comments

3. Abstract Wording

The sentence: “The outer kinetochore attaches to the microtubules and is named after three components KNL1C, MIS12C, and NDC80C (KMN).” could be revised for clarity. I recommend replacing “components” with either “sub-complexes” or “subunits,” as each of KNL1C, MIS12C, and NDC80C represents a multi-protein assembly.

> This is corrected in the revised manuscript.

4. Introduction Improvements

The introduction would benefit from a more detailed overview of the plant kinetochore structure. Including recent review articles (Xie Y, Wang M, Mo B, Liang C. Plant kinetochore complex: composition, function, and regulation. *Front Plant Sci.* 2024 Oct 10;15:1467236. doi: 10.3389/fpls.2024.1467236 ; Kozgunova E. Recent advances in plant kinetochore research. *Front Cell Dev Biol.* 2025 Jan 22;12:1510019. doi: 10.3389/fcell.2024.1510019) specific to plant kinetochore organization would enhance the context and strengthen the rationale for the study.

> We have extended the introduction and cited these two reviews.

5. BIN4 and Non-Canonical Interactions of KMN complex

The identification of proteins not directly implicated in cell division, particularly BIN4, is an interesting finding. BIN4, although distantly related to CENP-U, has not been associated with centromere or kinetochore functions in plants. Its co-precipitation with KMN components raises intriguing questions about potential indirect or novel interactions. I encourage the authors to mention this explicitly in the manuscript and offer possible interpretations, even if speculative, as this could inspire future functional investigations.

>This is an Interesting idea that is included in the revised manuscript.

Previous studies suggest that certain components of the Mis12 complex, such as Nnf1 (Allipra, S., Anirudhan, K., Shivanandan, S., Raghunathan, A. and Maruthachalam, R. (2022), The kinetochore protein NNF1 has a moonlighting role in the vegetative development of *Arabidopsis thaliana*. *Plant J*, 109: 1064-1085. <https://doi.org/10.1111/tpj.15614>), may play roles beyond their canonical kinetochore functions. In light of this, I am curious whether any of the proteins identified through the pull-down assay—particularly those not directly involved in cell division—might provide insights into potential non-kinetochore roles of the KMN complex or its interacting partners. Exploring or at least acknowledging this possibility could enrich the discussion and broaden the implications of the findings.

>This possibility was added to the revised manuscript.